# West Antarctic ice retreat and paleoceanography in the Amundsen Sea in the warm early Pliocene

Sandra Passchier [1] ✉, Claus-Dieter Hillenbrand [2], Sidney Hemming [3], Werner Ehrmann[4], Thomas Frederichs [5], Steve M. Bohaty[6], Ronald Leon[1], Olga Libman-Roshal [1], Lisbeth Mino-Moreira [1], Karsten Gohl [7] & Julia Wellner[8]

Mass loss from polar ice sheets is poorly constrained in estimates of future global sea-level rise. Today, the marine-based West Antarctic Ice Sheet is losing mass at an accelerating rate, most notably in the Thwaites and Pine Island glacier drainage basins. Early Pliocene surface temperatures were about 4 °C warmer than preindustrial and maximum sea level stood ~20 m above present. Using data from a sediment archive on the Amundsen Sea continental rise, we investigate the impact of prolonged Pliocene ocean warmth on the ice-sheet−ocean system. We show that, in contrast to today, during peak ocean warming ~4.6 – 4.5 Ma, terrigenous muds accumulated rapidly under a weak bottom current regime after spill-over of dense shelf water with high suspended load down to the rise. From sediment provenance data we infer major retreat of the Thwaites Glacier system at ~4.4 Ma several hundreds of km inland from its present grounding line position, highlighting the potential for major Earth System changes under prolonged future warming.

Over the past three decades, ice loss from the West Antarctic Ice Sheet (WAIS) has been a major contributor to sea-level rise. Over a 25-year period, the rate of mass loss increased five-fold as a result of dynamic thinning and grounding line retreat of glaciers draining into the Amundsen Sea Embayment, such as Pine Island and Thwaites glaciers[1–3]. During modern interglacial conditions, glaciers draining into the Amundsen Sea Embayment lose most of their mass through high rates of ocean-driven sub-ice-shelf melt and dynamic thinning upstream, in contrast to other parts of the Antarctic ice sheet, especially in East Antarctica, where warm deep water is upwelling further offshore from the slope and where iceberg calving dominates as a mechanism of mass loss[4–7]. Interglacial melt rates are primarily driven by the temperature and flux of Circumpolar Deep Water (CDW) as it interacts with ice shelves and grounding zones. Upwelling of CDW along the continental slope and inflow into cross-shelf troughs occurs as a result of a complex interplay of wind-driven and density-driven circulation, which, in turn, is impacted by meltwater release from the ice shelves[8–11].

The Amundsen Sea drainage sector of the WAIS holds a 125 cm sea level equivalent[12]. Given that the drainage basins in this sector, especially that of the Thwaites Glacier, is largely based below sea level on a reverse bed slope, current melting and mass loss there may lead to increasing rates of retreat[13] and attendant increase in sea-level rise. Recent meltwater deposits found on the Amundsen Sea continental

[1]Earth and Environmental Studies, Montclair State University, Center for Environmental and Life Sciences Rm 220, 1 Normal Ave, Montclair, NJ, USA. [2]British Antarctic Survey, High Cross, Madingley Road, Cambridge, UK. [3]Lamont-Doherty Earth Observatory, Columbia University, Palisades, NY, USA. [4]Institute for Earth System Science and Remote Sensing, University of Leipzig, Leipzig, Germany. [5]MARUM – Center for Marine Environmental Sciences, University of Bremen, Bremen, Germany. [6]Institute of Earth Sciences, Heidelberg University, Heidelberg, Germany. [7]Alfred Wegener Institute Helmholtz-Centre for Polar and Marine Research, Bremerhaven, Germany. [8]Earth & Atmospheric Sciences, University of Houston, Houston, TX, USA. ✉e-mail: passchiers@montclair.edu

shelf are interpreted as a signature of enhanced subglacial melt potentially contributing to fast retreat[14,15]. Ice-sheet dynamics are influenced by subglacial meltwater production and its routing at the ice-sheet bed[16] and warming may enhance surface melt and its drainage[17]. Periodic loss of ice shelves on the warmer Antarctic Peninsula has shown a strong dynamic response of ice flow, faster flow and increased sediment entrainment by the ice, with a possible role for meltwater percolation to the bed[18,19].

How these processes involving ocean−ice interactions play out over longer timescales than the observational record and in higher-than-today temperatures is poorly known and contributes to increased uncertainties on how the Antarctic system will evolve under the current climate changes. Whereas high-latitude feedbacks can result in non-linear responses to warming in the ice-sheet-atmosphere-ocean system, the effects may be different depending on local geological boundary conditions[20]. Ice-sheet modeling based on post-Last Glacial Maximum (ca. 19−23 ka) data from subglacial sediments in the Ross Sea sector of the WAIS suggested that uplift due to glacio-isostatic adjustment (GIA) created a prograde slope at the WAIS bed and, thus, allowed the grounding line to re-advance after initial deglacial retreat, whereas in the Amundsen Sea sector modeled retreat was steady without re-advance[21]. The variability in ice-sheet response points to the need to assess the regional context of ice-sheet and climate tipping points under warmer climate states than the dominant "icehouse" of the past 3.3 Ma[22], including different mechanisms of ice mass loss and how it was partitioned between East and West Antarctica[23,24].

During the early Pliocene, ~5.2 to 3.6 Ma, global surface temperatures were ~4 °C higher than pre-industrial under similar atmospheric greenhouse gas concentrations as today[25]. Sea surface temperatures and $pCO_2$ reached the highest levels of the past 5 Myrs during the early Pliocene[26,27], making it an excellent analogue for future evolution of the climate system. Reconstructed sea level high stands from the early Pliocene exceed 20 m above present[28] and require contributions from all major ice sheets, including the WAIS. A record of sea surface temperature (SST) variations based on Mg/Ca ratios of planktonic foraminifera at Deep-Sea Drilling Project (DSDP) Site 590 in the SW Pacific Ocean, can be used as a proxy for Southern Ocean subsurface ocean temperature variations[25,27], and shows that peak subsurface warming occurred in the early Pliocene. DSDP Site 590 lies at intermediate water depth and is influenced by low salinity and high nutrient Subantarctic Mode Water (SAMW) and Antarctic Intermediate Water (AAIW) that originate directly north of and within the Southern Ocean's Polar Frontal Zone[27]. Early Pliocene warming of the Southern Ocean was also noted in temperature reconstructions from the Ross Sea and Prydz Bay sectors that are based on silicoflagellates[29,30] and $TEX^L_{86}$ biomarker data from the Ross Sea shelf[31]. Geological archives show that the Antarctic ice sheet retreated from the Ross Sea and Wilkes Subglacial Basin with enhanced iceberg discharge in response to early Pliocene warming[32,33]. On the Antarctic Peninsula continental margin, prograding sequences on the outer shelf and upper slope and lithofacies in drill cores off Marguerite Trough show evidence for repeated late Miocene–early Pliocene advances and retreats of grounded ice streams across the shelf[34,35]. Coeval terrestrial evidence from the Antarctic Peninsula is interpreted in the context of a thinner wet-based ice sheet with several periods of ice retreat in the early Pliocene[36].

In the light of modern major ice loss in the Amundsen Sea drainage sector of WAIS, we targeted a sedimentary paleoarchive offshore from this sector to reconstruct ice-sheet variability through the warm early Pliocene. Here we use the sedimentological, geochemical and clay mineralogical data collected on the drillcores from the Amundsen Sea continental rise to determine the extent of WAIS retreat during the early Pliocene warm period, and to characterize the mechanisms and impacts of this ice loss. The seismic stratigraphy of the eastern Amundsen Sea outer continental shelf shows evidence of extensive progradation of ~80 km in the early Pliocene. However, shelf progradation became more variable and generally less extensive through time with evidence of buried grounding zone wedges (bGZWs) in the overlying aggradational intervals[37]. GZWs form at the grounding line under an ice shelf either when the ice-sheet terminus has reached its maximum extent or during short-term pauses that interrupt a time of general retreat. The bGZW in the aggradational Pliocene strata on the Amundsen Sea shelf document grounded ice advances to mid-to-outer shelf positions. Notably, these GZWs are buried under continuous, thick sediment drapes resulting from substantial periods (~300 kyr), during which grounded ice did not re-advance across the shelf and resided in a retreated position[37].

Although Pliocene-age sediments were not recovered in the few currently available short drill cores from the Amundsen Sea shelf[38], International Ocean Discovery Program (IODP) Expedition 379 Sites U1532 and U1533 recovered a hundreds of meters thick sequence of Pliocene sediments from a drift on the Amundsen Sea continental rise[39,40] (Fig. 1). Sites U1532 and U1533 were drilled in ~3900 − 4200 m water depth, ~250 km from the shelf break, into the Resolution Drift, which is located adjacent to a submarine channel (Fig. 1). Site U1533 was drilled on the western flank of the Resolution Drift directly to the east of the deep-sea channel[37,40]. The channel is connected to the eastern Amundsen Sea continental slope, offshore from the mouth of the Pine Island-Thwaites paleo-ice stream trough. Site U1532 was drilled near the crest of the same drift[39] (Fig. 1). The sites yielded Pliocene sediment archives of unprecedented resolution due to up to 90% core recovery and sedimentation rates up to dm/kyr, with inferred orbital-scale cyclical changes in sedimentation. The Pliocene sediments recovered at sites U1532 and U1533 consist of dark grey laminated silty clays interbedded with massive and bioturbated greenish grey silty clays with variable bioturbation and gravel content (Fig. 2). In preliminary shipboard analysis, four to five lithofacies were distinguished in the Pliocene drift sediments at both sites. These shipboard litho-facies classifications were based on descriptions and interpretations of representative examples of dm-scale intervals of sedimentary structures or clast-rich beds[39,40].

In this study we embed these dm-scale interpretations into a lithofacies assemblage scheme applied to the deposition, over time, of the hundreds of meters thick lower-to-mid Pliocene sequence over multiple glacial-interglacial cycles[41]. We investigate the offshore supply of glaciogenic detritus through the early Pliocene within a chronological framework of orbital-scale cyclical changes in sedimentation. Using facies models in conjunction with grain-size measurements and detrital provenance, we aim to reconstruct the connections between source-to-sink supply of glaciogenic detritus and WAIS dynamics. Terrigenous sedimentation on the Antarctic continental rise is governed by glacial processes on land as well as the continental shelf that produce terrigenous detritus, downslope transport of this material to the continental slope and rise, and bottom current reworking[41–43]. Over glacial-interglacial cycles, the depositional signatures of ice-sheet and ice-shelf processes in deep-marine sediment drifts are expressed in muddy turbidites, hyperpycnal plumites, contourites and sand and gravel-rich ice-rafted debris (IRD) beds[41–44]. Besides IRD, glacial rock flour, which is silt-sized and supplied to the glacio-marine environment, can be distributed in hyperpycnal flows, where suspended particles are redeposited by bottom currents of variable strength.

We deploy a multiproxy approach with bulk trace and Rare Earth Element (REE) geochemistry, including X-ray fluorescence (XRF) scanner data, clay mineralogy, and $^{40}Ar/^{39}Ar$ geochronology of medium-coarse sand fractions to track the provenance of the fine-grained drift sediments and the coarse-grained ice-rafted component. Grain-size data collected for this study is used to calculate the sortable silt mean size ($\overline{SS}$), which can be interpreted as a proxy for bottom-current strength[45]. The data collected and analyzed post-cruise allows us to build an early to mid-Pliocene time series of changes in the

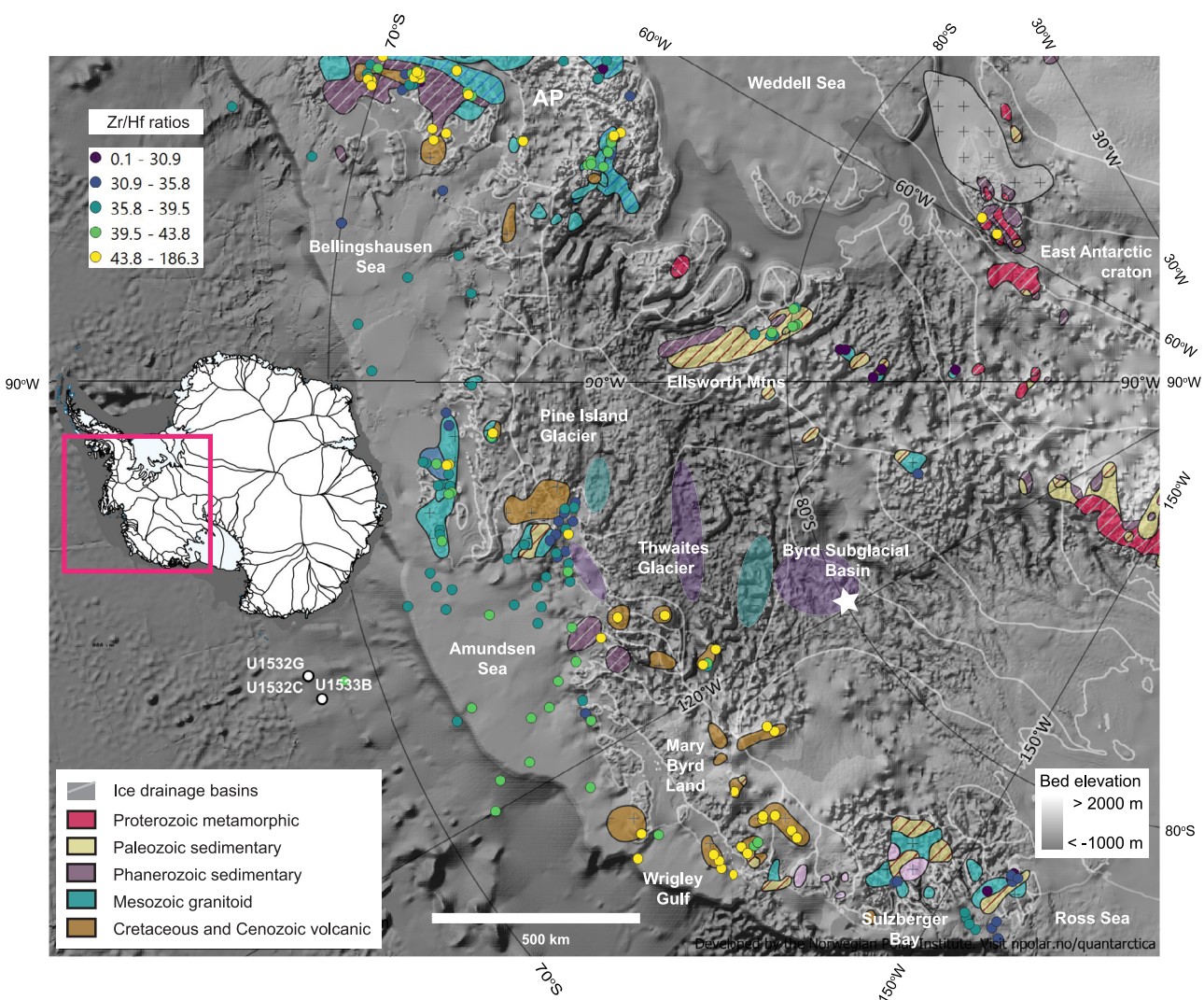

**Fig. 1 | Locations of IODP Expedition 379 drill sites (U1532C, U1532G, U1533B) and the Pine Island Glacier and Thwaites Glacier drainage basins.** White lines delineate the present-day glacial drainage basins. The Antarctic whole rock geochemical data was extracted from the database of Sanchez et al. [73] and references therein (see Supplementary References). The offshore geochemical data is from Simões Pereira et al. [54,55] The star marks the location of the Byrd Ice Core with the upstream subglacial distribution of kaolinite-rich Phanerozoic sediments based on Marschalek et al. [60] Subglacial geology of Thwaites Glacier is after Jordan et al. [48] The base map was produced using the geospatial data compilation Quantarctica[97]. The geological map compilation is from Tingey[98]. The shaded relief map is based on BEDMAP2[99].

sources of suspended and ice-rafted detritus from the Thwaites and Pine Island Glacier drainage systems that inform the extent in changes in grounding line position within the context of the changing local paleoceanography.

The geology of West Antarctica is the product of multiple tectonic collisions with accompanying magmatic events[20,46–48]. The Eastern Palmer Land Shear Zone separates the Paleozoic Gondwana terranes of the Eastern Domain from terranes amalgamated during Triassic–Jurassic events (Central Domain)[49]; the Western Domain, adjacent to the Bellingshausen and Amundsen seas, consists of terranes affected by Cretaceous accretionary processes, rifting, and magmatic events (Fig. 1). Paleozoic metasediments and granitoids related to the active margin of Gondwana dated to ~375 to ~250 Ma are found in Marie Byrd Land, whose ice-sheet cover is partly drained through Thwaites Glacier[50]. In contrast, granitoids from the Thurston Island block ( ~240 − 220 Ma)[51], near Pine Island Glacier ( ~180 − 170 Ma)[52] and from the Ellsworth-Whitmore Mountains (250 − 170 Ma)[53] have Triassic–Jurassic ages. These plutons were emplaced during a younger phase of subduction along the margin of Zealandia[51]. The boundary of the Central Domain with Triassic–Jurassic arc-volcanic intrusives and the

Marie Byrd Land province, which lacks evidence of magmatic activity of this age, runs under Thwaites Glacier[52,54].

In the mid-Cretaceous ~110 − 100 Ma, flare-up arc volcanism affected Thurston Island and the Antarctic Peninsula[47,51]. In a different tectonic setting, magmatic phases related to the early stages of Cretaceous rifting of Zealandia and West Antarctica are observed in Marie Byrd Land ~100 − 95 Ma. Isolated highlands of felsic rocks with low magnetic susceptibility detected ~200 km landward of the grounding line under Thwaites Glacier may represent similar anorogenic granites[48] (Fig. 1). Mafic igneous rocks and sedimentary basins created during the later stages of Cretaceous-Cenozoic rifting are interpreted to dominate the subglacial geology of the Thwaites Glacier system[54,55].

Late Cretaceous and Cenozoic rifting created sedimentary basins, which are currently partially covered by the WAIS[56]. The relatively high kaolinite content of the clay mineral fraction of seafloor sediments in the Amundsen Sea is indicative of a sedimentary rock source[57–59]. Kaolinite concentrations of >30% are found in modern sediments near Thwaites Glacier, with lower kaolinite contents characterizing sediments with a Pine Island Glacier source[55]. Basal debris from the Byrd ice-core in the WAIS interior contains ~40% kaolinite and is inferred to

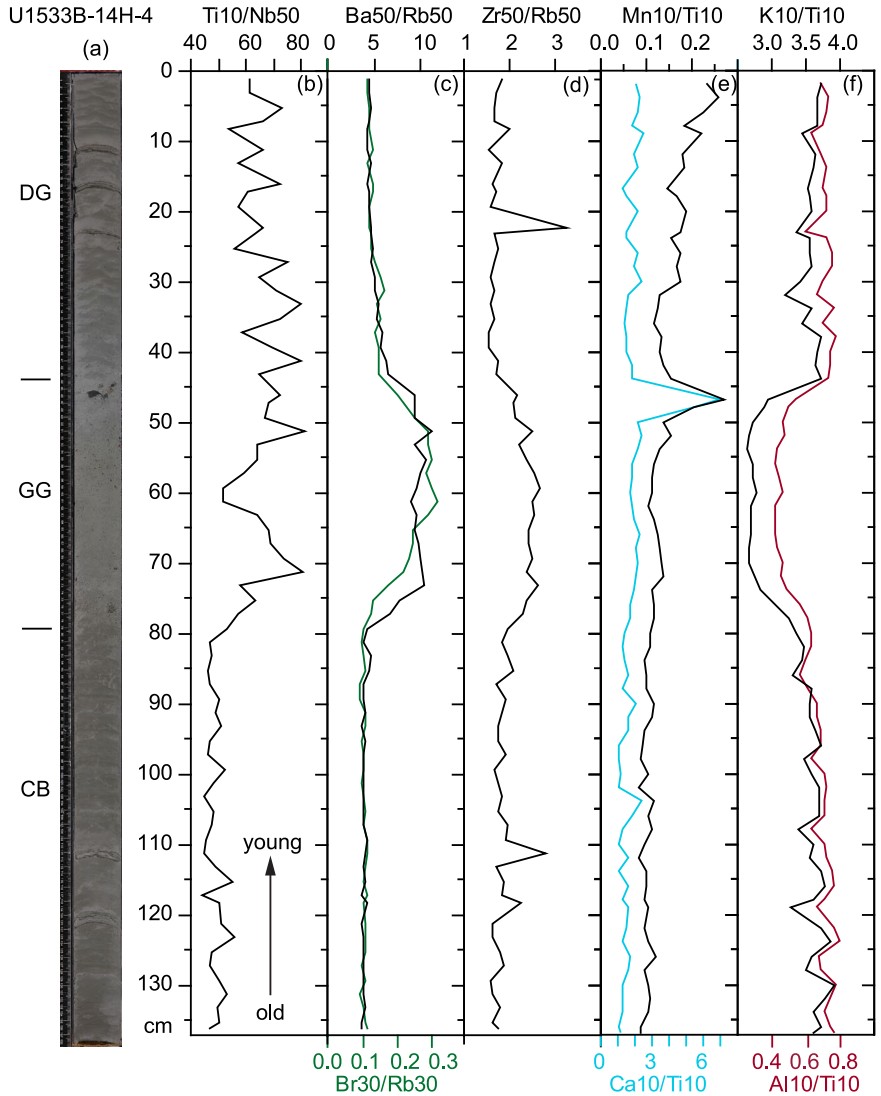

**Fig. 2 | Example of the sequence of a typical transition from a glacial (terrigenous facies) to deglacial (color-banded facies) and interglacial (greenish grey facies) conditions as seen from XRF scanning data and core image of U1533B-14H section 4.** Panel (**a**) core image with facies characterization from old to young and bottom to top: color-banded silty clay (CB) that is moderately bioturbated, mostly terrigenous with occasional siliceous fragments and rare outsized clasts, grading upward into greenish grey massive and bioturbated diatomaceous silty clay (GG) with outsized clasts, including pebbles, overlain by dark grey laminated silty clay (DG), mostly terrigenous with turbidites and rare outsized sand and gravel, (**b**) elevated Ti/Nb values in the greenish grey facies and the overlying laminated clays, (**c**) elevated Ba/Rb ratios in the greenish grey facies, (**d**) increases in Zr/Rb values in the greenish grey facies, with maxima in the other facies related to coarse-grained laminae, (**e**) maxima in Ca/Ti and Mn/Ti at the upward transition from greenish grey to dark grey laminated facies, and (**f**) lower K/Ti and Al/Ti ratios in the greenish grey facies, signaling a different provenance for the terrigenous fraction. Source data are provided as a Source Data file.

derive from the infill of subglacial sedimentary basins[60]. Kaolinite is also found in pre-Oligocene sediments or sedimentary rocks cropping out on the Amundsen Sea inner shelf[57–59].

Here we show that early Pliocene sedimentation on the Amundsen Sea continental rise was sourced within the Amundsen Sea ice drainage sector and strongly coupled to advance and retreat of Thwaites Glacier. Following a period of rapid accumulation of meltwater plume deposits a temporary minimum in kaolinite content and Marie Byrd Land-type geochemical and $^{40}$Ar/$^{39}$Ar IRD provenance signatures in the glaciogenic supply is indicative of major retreat of Thwaites Glacier ~4.4 Ma several 100 s of km upstream of its present grounding zone. Biogenic and authigenic geochemical signatures and the sortable silt current proxy indicate that vigorous circulation gave way to slow-moving currents ~4.6–4.1 Ma, whereas meltwater plumes extended into deepwater more than 250 km from the shelf break. These results are significant as they highlight the potential impacts of major WAIS retreat after its ice-sheet inception during a prolonged warm period

with surface temperatures only a few degrees above those of the present day.

## Results

### Sedimentary facies

Based on shipboard visual core and smear-slide descriptions, augmented with post-cruise image analysis and new grain-size data acquired post-cruise for this study, three main lithofacies are distinguished in the early to mid-Pliocene section at sites U1532 and U1533: 1) terrigenous dark grey laminated silty clay with or without sand interbeds; 2) terrigenous interlaminated silty clay with reddish color banding, and 3) greenish grey silty clay with variable intensity of bioturbation, contents of gravel and biogenic silica components. Using XRF-scanner data and detailed analysis of a* track data carried out for this study post-cruise, these facies are here placed in a cyclostratigraphic context. These three facies typically occur in an interbedded sequence with the color-banded facies transitioning upward into the

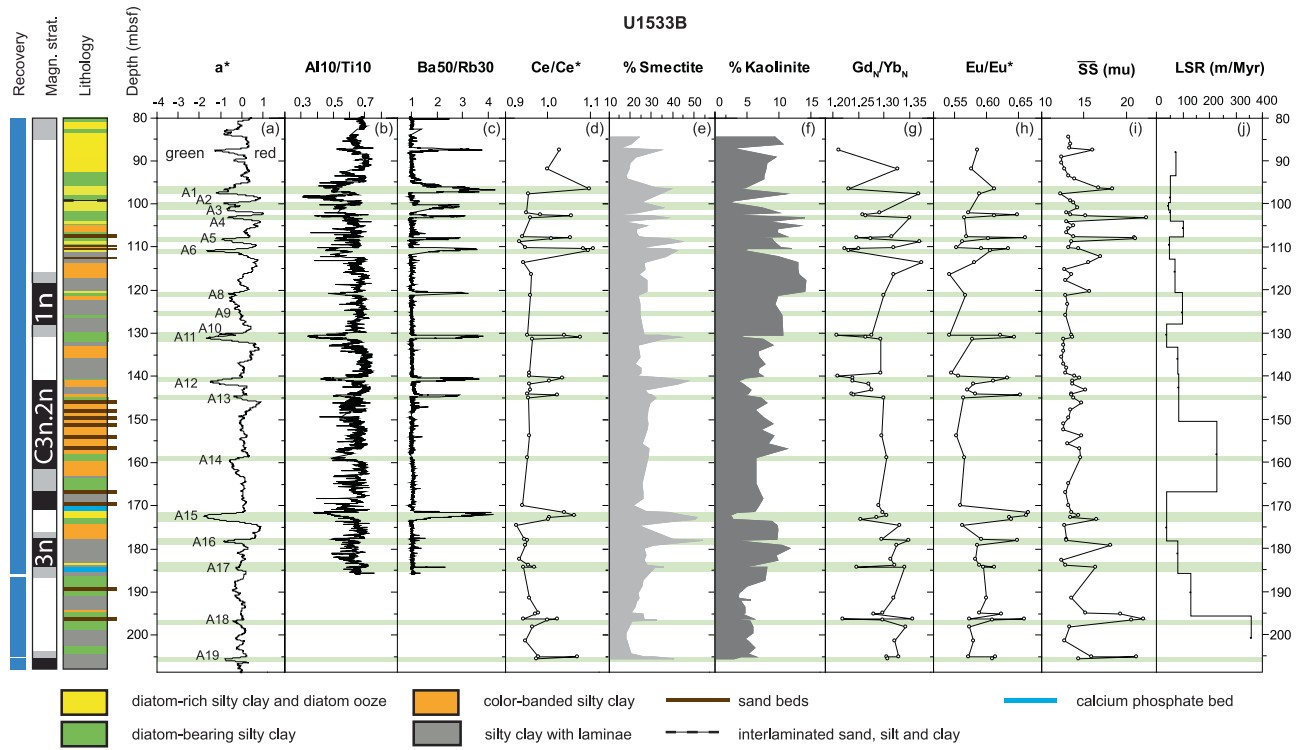

**Fig. 3 | Early Pliocene glacial-interglacial cyclicity in sediment composition between 80 and 210 mbsf and sediment provenance changes in Hole U1533B.** Panel (**a**) a* color photospectrometry record of Site U1533, with low values indicating green and high values red hues[40], (**b**) Al/Ti ratios derived from XRF scanner data, (**c**) Ba/Rb ratios derived from XRF scanner data, (**d**) Cerium anomaly values derived from bulk sample ICP-MS analysis, (**e**) % Smectite and (**f**) % Kaolinite in the clay fraction <2 μm, (**g**) chondrite-normalized Gd/Yb ratios and (**h**) Europium anomaly, derived from bulk sample ICP-MS analysis, (**i**) mean sortable silt grain size, and (**j**) linear sedimentation rates (LSR). Magn. strat. = shipboard interpreted

magnetic polarity and chronostratigraphy. Greenish grey glacial retreat facies have high Ba/Rb, smectite and Cerium anomaly values, and Al/Ti ratios. Mean sortable silt grain size ($\overline{SS}$) maxima are higher in the upper and lower portions of the displayed interval and contrast with the linear sedimentation rates (LSR). Kaolinite contents show variations in clay mineral provenance, with higher kaolinite contents indicating more detrital supply from the Thwaites Glacier basin. Higher $Gd_N/Yb_N$ ratios of the bulk sediment are also indicative of the Thwaites Glacier supply. The Europium anomaly is exceptionally low for the interval with common color-banded facies between ~110 and 170 mbsf. Source data are provided as a Source Data file.

greenish grey gravel bearing facies, which is typically overlain with a sharp contact or transition by the dark grey laminated facies (Fig. 2; Supplementary Fig. 1). The dark grey, terrigenous silty clays contain rare dispersed sand or gravel and show normally graded, fanning, deformed, or wavy silt laminations, ripple cross-lamination, and silt lenses, primarily in the lower Pliocene section at both sites[39,40]. Color-banded silty clays are thinly laminated and locally moderately bioturbated with occasional fragments of siliceous microfossils and rare dispersed sand and gravel. Laminations occur by color and do not signify visible grain-size differences. A reddish hue is common. Sets of thin sharp-based sand beds occur throughout this facies. The greenish grey facies consists of massive or bioturbated diatomaceous silty clay and generally exhibits an upward transition from a bioturbated mud with dispersed gravel in the lower part of the unit to more gravel-rich mud at the top, which locally exhibits stratification and a higher biogenic content (Fig. 2).

Grain-size distributions for the greenish grey facies are most variable with modes ranging from clay to coarse silt (Supplementary Fig. 2). Dark grey laminated silty clays also show variable silt modes. In contrast, muds of the color banded facies typically consist of more uniform fine silt (Supplementary Fig. 2). A strong correlation between sortable silt and sortable silt mean size is found for the silty clays, indicating a variable bottom current influence. Five samples of gravel-bearing greenish grey lithology and one from the dark grey laminated facies exhibit a uniform fine sand mode with a fine tail. These samples were excluded from the sortable silt calculations.

Bulk geochemical data also reveal clear changes between the facies. XRF-scanner derived Ti/Nb ratios change abruptly near the

lower boundary of greenish grey units, and in contrast to Al/Ti ratios, a very gradual decrease is observed near the upper boundary of greenish grey units (Fig. 2). Increases in Ca/Ti and Mn/Ti ratios are found at the tops of greenish grey units. The facies variability is also noted in the down-core distributions of all datasets (Fig. 3). Color-banded intervals have the highest a* values near the tops of the intervals, indicating that they are the most reddish in color, whereas greenish grey units have the lowest a* values. Al/Ti ratios follow a cyclic pattern related to facies with minima in the greenish grey facies only. Greenish grey units have low a* values and higher Ba/Rb ratios and Ce/Ce* values than the laminated silty clays. Greenish grey units also have higher smectite contents, whereas the highest kaolinite percentages coincide with color-banded facies at Site U1533 (Fig. 3). The color-banded terrigenous facies occur primarily within cycles between ~180 and 110 mbsf at Site U1533 and are concentrated between ~170 and 140 mbsf (Fig. 3), whereas color-banding is not as well developed at Site U1532 (Supplementary Fig. 3), which is located near the crest of Resolution Drift, i.e., further away from the deep-sea channel. Similarly, kaolinite contents show large-scale changes over the depth of the drilled interval (Fig. 3). In Hole U1533B the kaolinite content displays a minimum at ~140 mbsf and increases (>12%) between ~130 and 110 mbsf (Fig. 3). $Gd_N/Yb_N$ ratios and kaolinite contents generally correlate well, except above ~100 mbsf. A similar pattern is observed in correlative strata at Site U1532 (Supplementary Fig. 3). The sediments of all facies become more diatom-rich in the upper parts of the early-mid Pliocene sections at both sites, albeit without a change in the terrigenous lithological expression of the facies cyclicity (Fig. 3, Supplementary Fig. 3).

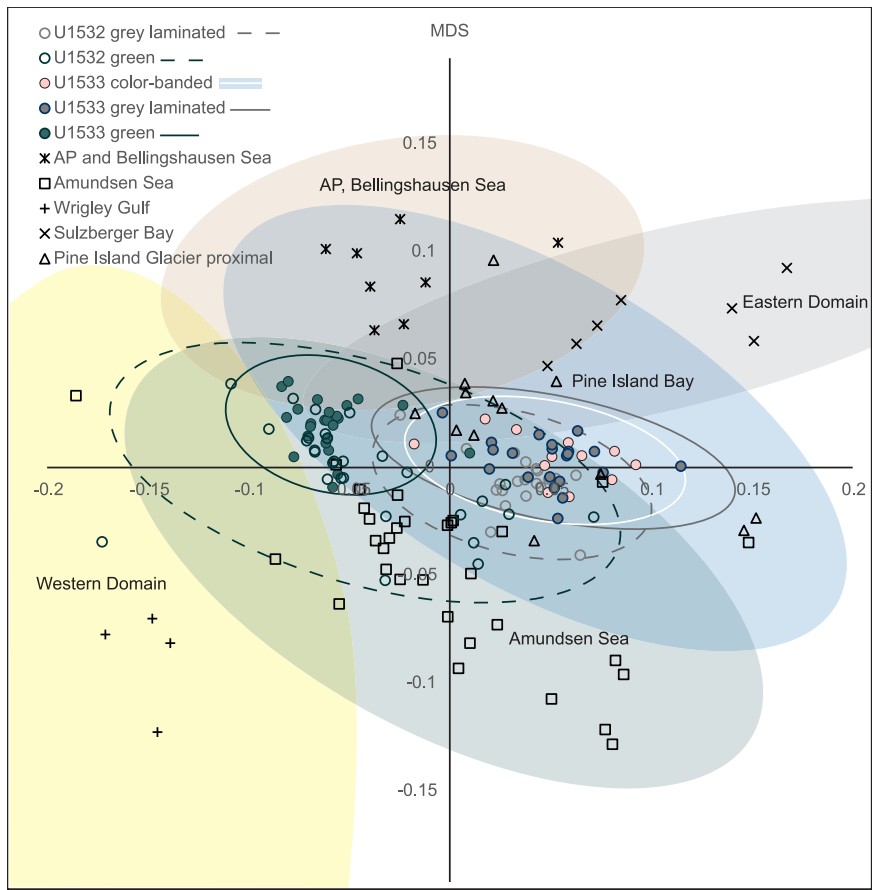

**Fig. 4 | Non-metric multidimensional scaling of detrital geochemical data.** Data from lithofacies at sites U1532 (colored line circles) and U1533 (solid colored circles), and seafloor surface mud fractions (< 63 µm) of sediments from the West Antarctic continental shelf[54,55] (black symbols) (AP: Antarctic Peninsula). Relative distance between data points reflects geochemical similarity, which is used to determine the provenance of the Amundsen Sea continental rise sediments. The elemental ratios used in the statistical analysis were $Gd_N/Yb_N$, La/Th, Sm/Zr, Th/Sc, Zr/Hf and the Eu/Eu*. Line ellipses represent 95% confidence limits with solid lines engulfing data for U1533 and dashed lines data for U1532. Colored ellipses without lines span 95% confidence limits for the continental shelf data of Simões Pereira et al.[54,55] Source data are provided as a Source Data file.

## Interpretation of dark grey laminated and color-banded facies

The dark grey laminated facies in the Pliocene sections at sites U1532 and U1533 exhibit sedimentary structures consistent with intermittent bottom-current processes, whereas current structures are not ubiquitous in the color-banded facies. Sharp-based sand beds in both the dark grey laminated and color-banded facies are interpreted as turbidites[39,40]. Both the dark-grey laminated and the color-banded facies are interpreted as a deposit from downslope sediment transport with bottom-current reworking. However, the grain-size distributions for the color-banded sediments (between thin sharp-based sand beds) at Site U1533 exhibit a lower sortable silt mean size than the other facies, confirming deposition of suspended material under more limited bottom-current strength (Supplementary Fig. 2).

Low Eu/Eu* values of the laminated and color-banded facies at Sites U1532 and U1533 (Fig. 3) are typical of a felsic igneous source and point to a supply of detritus from granitoids within the Pine Island Glacier basin, which is characterized with Eu/Eu* values as low as 0.51 – 0.59 in the mud fraction of modern sediments[55] (Supplementary Fig. 4). The multidimensional scaling of trace elements shows that the provenance of the dark grey laminated silty clay and especially the color-banded silty clay facies is most similar to modern sediment in proximity to Pine Island Glacier (Fig. 4). In addition, correlative increases in bulk sediment $Gd_N/Yb_N$ ratios and kaolinite contents indicate periodic increases in glaciogenic detritus supply from Thwaites Glacier and ice draining into the western Amundsen Sea embayment[54–56]. Kaolinite-bearing sediments or sedimentary rocks are present within the Amundsen Sea inner shelf and underneath WAIS in the Byrd Subglacial basin[57–60] and higher $Gd_N/Yb_N$ values are indicative of the more mafic and younger, less evolved, magmatic signature of the West Antarctic Rift System/Marie Byrd Land province (Supplementary Fig. 4).

While sedimentary structures, such as ripple cross-lamination, silt and fine-sand lenses, and fanning laminations, in the dark-grey laminated facies exhibit evidence of bottom current reworking, the color-banded facies lacks these structures and is primarily composed of fine-grained color-laminated mud with intermittent thin sharp-based sand beds interpreted as distal turbidites. We interpret the color-laminated facies at Site U1533 as a depositional signature of fine-grained suspended sediment plumes originating near the grounding zones in the Amundsen Sea Embayment, either from subglacial channelized flow or ice-shelf melt or both, that were transported to the continental rise as hyperpycnal plumes and flows. Hyperpycnal flows can transport fine-grained detritus to the slope and rise through spill-over from the shelf[42,61,62]. These deposits are rare on the present-day Antarctic continental margin, but meltwater deposits (plumites) have been described offshore from more temperate glacial margins, where they extend several 100 s of kilometers out from a former ice-stream margin, and where suspended sediment is interpreted to have reached the continental slope and rise as hyperpycnal plumes entrained within low-intensity geostrophic flows[42,44,61,62].

### Interpretation of greenish grey facies

The greenish grey facies with bioturbation, a larger biogenic component, and dispersed sand and gravel, are interpreted as the product of hemipelagic deposition with iceberg rafting at the onsets of glacial retreat. Greenish grey biogenic silica-bearing facies are distinctly different in geochemical and clay mineralogical composition from the terrigenous laminated and color-banded facies (Fig. 2). In contrast to the grey laminated and color-banded facies, the bulk sediment of the greenish grey facies does not exhibit a strong eastern Amundsen Sea Embayment or Pine Island Glacier detrital provenance signal (Fig. 4). Low Al/Ti ratios characterize the greenish grey units at Site U1533 and because Al scavenging by diatoms would elevate Al/Ti ratios and not decrease it, the change in Al/Ti ratios is most likely a detrital signal. High Ti content is found in South Pacific dust. However, Antarctic ice cores have higher dust content in Pleistocene glacials[63], while no clear dust signal is detectable in sediments from the Amundsen Sea continental rise during the present interglacial[64]. Thus, the low Al/Ti ratios more likely indicate reduced terrigenous supply from the WAIS during Pliocene interglacials. High Ti/Nb ratios above the bases of the greenish grey facies (Fig. 2) are a strong indicator of a shift towards mafic igneous source rock contributions at glacial terminations; this elemental ratio is not likely affected by weathering, diagenesis or sediment sorting[65]. Enrichment in smectite and high Eu/Eu* values in the greenish grey facies (Fig. 3) are consistent with a mafic volcanic source. Detritus with a mafic igneous provenance signature in the Amundsen Sea may be supplied from local West Antarctic volcanic sources[54,57], but an offshore marine source is also possible[58], which at least for some of the smectite has recently been confirmed by detailed investigations of smectite chemistry[66]. High Eu/Eu* ratios >0.6 in the greenish grey facies (Fig. 3) are consistent with a detrital mud fraction derived from an offshore South Pacific source, as modern silt and clay-sized detritus in the deep easternmost Pacific sector has Eu/Eu* values ranging from 0.63 to 0.69 (data from Wengler et al.[64]).

Elevated Mn/Ti ratios at the top of the greenish grey facies and within the transition to the overlying grey laminated facies point to generally well-oxygenated bottom water (Fig. 2). Low terrigenous supply under well-oxygenated bottom water conditions is a favorable setting for the formation and accumulation of authigenic minerals, which is consistent with the high Ce/Ce* values (positive Ce-anomalies) and the elevated Ba/Rb ratios attributed to deposition of water-column derived barium sulphate (barite) under high productivity conditions (Fig. 3). ICP-MS results on a discrete sample at the top of the greenish grey unit (A15) in Core U1533B-17H at ~171.5 mbsf show that the ripple-laminated sands at that level are interbedded with indurated calcium, manganese and phosphorous-rich sediment (Fig. 3; Supplementary Fig. 1). A deformed carbonate-fluorapatite concretion or bed with high Ca/Ti ratio was observed shipboard via XRD analysis in Core U1533B-18H-6 at 185.6 mbsf[40]. We interpret these carbonate-fluorapatite beds as condensed sections resulting from a decrease in terrigenous supply, biosiliceous bloom activity during interglacials, and bottom-current reworking in ocean frontal zones. Diatoms are efficient fixators of P and are involved in the generation of most marine calcium phosphate deposits, in which Ca can be replaced by Ce under well-oxygenated bottom water conditions[67,68].

### Correlation of greenish grey units and refined age model

In the lower Pliocene sections of sites U1532 and U1533, greenish grey hemipelagic beds were correlated using the shipboard magnetostratigraphy, the a* record, abundance of biogenic components, and magnetic susceptibility (Fig. 5). Using this approach we were able to correlate ~20 early Pliocene greenish grey beds between the two sites (Table 1). We were also able to tie a distinct bed sequence at U1532C-7F-3, 55 cm (205.37 mbsf) and U1533B-9H-6, 0 cm (99.31 mbsf), dated to 3.92 and 3.91 Ma, respectively, applying the individual linear age models for the two sites. A few greenish grey beds were only present or

recovered at one site. The sequence of facies in each cycle drilled at sites U1532 and U1533 is indicative of the transient behavior of the ice-sheet ocean system in the Amundsen Sea sector over Pliocene glacial-interglacial cycles. Following the facies model (Fig. 6), the greenish grey beds are interpreted to represent interglacial stages of deposition. The tops or a* minima of correlative greenish grey bed sequences on the Amundsen Sea continental rise, labeled A1 to A23 (Fig. 5), can be tied to glacial-interglacial transitions derived from the LR04 benthic stack[69] (Table 1). This approach was used to attribute ages to the A1 – A23 greenish grey units and to refine the age models for both sites through linear interpolation between glacial-interglacial age tie points (Supplementary Fig. 5).

### $^{40}Ar/^{39}Ar$ dating of IRD in greenish grey facies

The greenish grey facies contains variable concentrations of dispersed sand and gravel grains interpreted as IRD. $^{40}Ar/^{39}Ar$ dating of 228 out of 230 individual ice-rafted hornblende, biotite and other mica grains (>250 μm) in the greenish grey units at both sites U1532 and U1533 revealed prevalence of Triassic to Cretaceous ages (Fig. 7; Supplementary Fig. 6), with two grains exhibiting negative ages removed from the dataset. The main modes are concentrated at ~93 and ~170 Ma. The ~170 Ma mode overlaps with the ages of Triassic–Jurassic intrusives known from the Ellsworth-Whitmore and Thurston Island blocks, including ~180 – 170 Ma granitoids near the downstream end of the Pine Island Glacier catchment[51–53]. In contrast, the ~93 Ma age is not characteristic for recent IRD sourced from the Amundsen Sea sector and the nearest known outcrops and offshore sediments recording these ages are found along the west side of the Antarctic Peninsula[54]. Therefore, the ~93 Ma old grain population could represent IRD delivery from icebergs drifting with the westward flowing Antarctic coastal current from sources on the Antarctic Peninsula and adjacent terranes (Fig. 1). Ages of IRD are not uniform and vary between intervals (Fig. 8). Notably, an IRD grain population with dominant Triassic–Jurassic ages is observed in the greenish grey units between 140 and 145 mbsf in Hole U1533B and between 290 and 320 mbsf in Hole U1532C (IRD-bearing beds A12 – A13).

## Discussion

The facies distribution and compositional changes in the Pliocene stratigraphy drilled at Sites U1532 and U1533 in the Amundsen Sea sector reflect three phases of marine ice-sheet and ocean circulation regimes centered around an ice-sheet minimum between ~4.6 and 4.3 Ma. The lower sequence in the seismic stratigraphy off the eastern Amundsen Sea continental shelf shows evidence of extensive progradation in the late Miocene to earliest Pliocene, which is interpreted as a record of ice advances to the continental shelf break[37]. Throughout the lower Pliocene sequence the thick dark grey laminated sediments at Sites U1532 and U1533 are characterized by a provenance signature typical of the Amundsen Sea hinterland, indicating down-slope sediment delivery during glacials (Fig. 4), and sedimentary structures[39,40] reflecting bottom-current reworking on the continental rise (Fig. 3). Ocean modeling shows that Pliocene ice advance in the Pacific sector of Antarctica increased the wind stress at the surface that drives the east-ward flowing Antarctic Circumpolar Current (ACC), the westward flowing Antarctic coastal countercurrent, and the overturning circulation[70].

Interglacial units A19 – A18, older than ~4.6 Ma exhibit La/Th ratios close to upper crustal averages (Fig. 9) that can be attributed to the presence of biogenic apatite in marine sediments[71], and resembling those of dust deposited in the modern Pacific Ocean north of the Antarctic Polar Front[64]. Higher La/Th values are also consistent with a source in the Cretaceous–Cenozoic central and western domain of West Antarctica, indicating glacial coverage and minimal WAIS retreat (Fig. 1 and Supplementary Fig. 4). Contents of terrigenous sand are low in interglacial units A19 – A15 (Fig. 9), implying, with decreasing linear

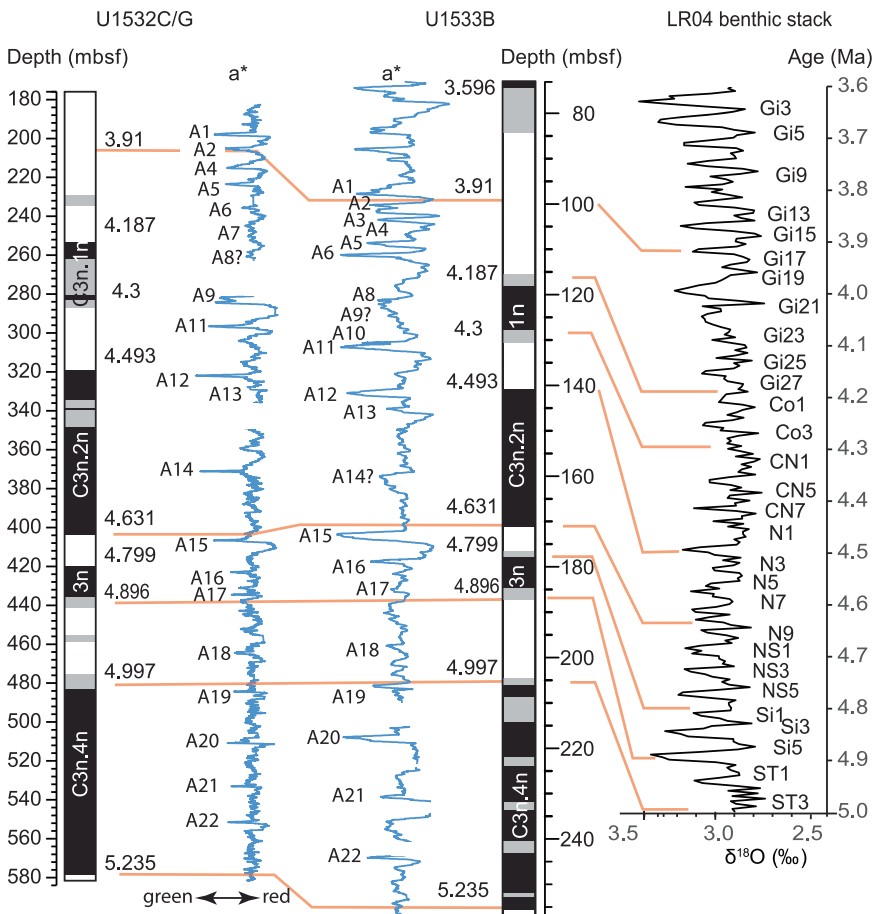

**Fig. 5 | Correlation of the a\* color photospectrometry records of sites U1532 (holes U1532C and U1532G) and U1533 (Hole U1533B) vs.** interpreted magnetic polarity (grey shading: uncertain polarity)[39,40] and the LR04 benthic stack[69,95].

Greenish grey facies stand out as low, green, values on the a\* (red-green ratio). Source data are provided as a Source Data file.

sedimentation rates between ~5.0 and 4.6 Ma, that millennial-scale IRD mass accumulation rates were lower in these interglacials than in younger interglacials. The $^{40}Ar/^{39}Ar$ ages of IRD grains in interglacial units A19 − A15, are clustered around 88 − 90 Ma with a secondary mode at ~170 − 220 Ma (Fig. 8) most typical of sources on the Antarctic Peninsula and Thurston Island block[51,54,55]. These results suggest that the coastal current was the primary delivery mechanism for ice-rafted debris from a glaciated Antarctic Peninsula and Thurston Island before ~4.6 Ma. This interpretation is in agreement with the seismic evidence of prograding sequences on the Antarctic Peninsula outer continental shelf and sediment records, as well as terrestrial evidence, documenting early Pliocene advances of grounded ice to near the shelf break and periods of ice-sheet retreat[34–36].

In greenish grey interglacial facies older than ~4.6 Ma at Site U1533 higher sortable silt is observed, whereas smectite contents and Eu/Eu\* values of interglacials A16 and A15 are very high at both Sites U1532 and U1533, suggesting increased southward current-induced advection of detritus from the Pacific basin, and not from the Antarctic margin (Fig. 10). Prominent calcium phosphate deposits associated with ripple lamination at Site U1533 likely accumulated under low terrigenous sedimentation rates through biosiliceous bloom activity and bottom-current reworking in ocean frontal zones, where upwelling of relatively warm Pacific deep waters mixed with nutrient-rich and well-oxygenated Antarctic surface waters. The interaction between large ice sheets and warmer ocean currents led to phases of ice retreat, during which sediment from the Amundsen Sea sector did not reach the continental rise, such as during peak ocean warming at ~4.8 and 4.6 Ma (Fig. 10). Lamy et al.[72] suggested that increases in the intensity

of the ACC in the early Pliocene brought warmer water masses in contact with the WAIS. Our data show that, ~4.8 and 4.6 Ma, the ACC, with its Southern Front today being located at ca. 66°S directly to the north of the Expedition 379 sites[72], had either strengthened, widened, or shifted southward by 2.5° of latitudes to sites U1532 and U1533 during interglacials. Southward frontal displacements in the Southern Ocean during early Pliocene interglacials have previously been inferred for the Antarctic Pacific sector[72].

After ~4.6 Ma a major weakening in current strength, increase in linear sedimentation rates, and change in sediment provenance occurred at both sites ( ~160 mbsf in Site U1533 and ~370 mbsf at Site U1532; Fig. 3; Supplementary Fig. 3). The corresponding section has relatively thick beds of uniform color-laminated terrigenous facies at Site U1533 and is characterized by very high linear sedimentation rates and sets of sharp-based sand beds at both sites. An increase in detrital sand content in samples from the greenish grey interglacial facies, coincident with higher linear sedimentation rates, also suggests an increase in IRD accumulation rates at glacial terminations ~4.6 − 4.5 Ma (Fig. 9; A14 − A12). Both sites also show an abrupt upward increase in kaolinite under very high linear sedimentation rates, followed by a gradual decrease to a minimum of <5% kaolinite accompanied by a decrease in linear sedimentation rates (Fig. 10).

The abrupt increase in kaolinite, followed by a gradual decrease to very low kaolinite contents and low correlative $Gd_N/Yb_N$ ratios ~4.6 − 4.5 Ma in both glacials and interglacials may indicate, first, an advance of the Thwaites glacier system, followed by a decline to a lack of material from a Thwaites Glacier/Marie Byrd Land source ~4.5 − 4.4 Ma (Fig. 10). A proposed terrane boundary under Thwaites

**Table 1 | Correlation of interglacial units between Sites U1532, U1533 and the LR04 benthic stack**

| Green units A # | U1532 Depth (m) | U1532 Age Model (Ma) | U1533 Depth (m) | U1533 Age Model (Ma) | U1532-U1533 Age diff (kyr) | Exp379 Average age (Ma) | LR04 MIS | LR04 Age (ka) | Exp379_LR04 Age diff (kyr) |
|---|---|---|---|---|---|---|---|---|---|
| 1 | 197.47 | 3.88 | 95.60 | 3.85 | −26.40 | 3.86 | Gi13/Gi14 | 3862 | 1.88 |
| 1 | 198.84 | 3.88 | 97.38 | 3.88 | −7.27 | 3.88 | Gi13/Gi14 | 3862 | 18.98 |
| 2 | 202.73 | 3.91 | 98.19 | 3.89 | −16.56 | 3.90 | Gi15/Gi16 | 3912 | −14.24 |
| 2 | 205.71 | 3.92 | 99.87 | 3.91 | −7.80 | 3.92 | Gi15/Gi16 | 3912 | 6.54 |
| 3 |  |  | 101.06 | 3.93 |  | 3.93 | Gi17/Gi18 | 3939 | −6.53 |
| 4 | 213.36 | 3.96 | 102.51 | 3.95 | −10.37 | 3.96 | Gi19/Gi20 | 3978 | −18.62 |
| 4 | 215.77 | 3.98 | 103.11 | 3.96 | −14.65 | 3.97 | Gi19/Gi20 | 3978 | −7.49 |
| 5 | 223.24 | 4.02 | 107.76 | 4.03 | 13.88 | 4.03 | Gi21/Gi22 | 4029 | −3.10 |
| 5 | 223.96 | 4.02 | 108.23 | 4.04 | 16.96 | 4.03 | Gi21/Gi22 | 4029 | 2.41 |
| 6 |  |  | 110.20 | 4.07 |  | 4.07 | Gi23/Gi24 | 4085 | −15.60 |
| 6 |  |  | 110.81 | 4.08 |  | 4.08 | Gi23/Gi24 | 4085 | −6.46 |
| 7 | 253.81 | 4.19 |  |  |  | 4.19 | Gi27/Gi28 | 4192 | −4.82 |
| 8 |  |  | 120.58 | 4.21 |  | 4.21 | Co1/Co2 | 4232 | −18.13 |
| 8 |  |  | 120.78 | 4.22 |  | 4.22 | Co1/Co2 | 4232 | −16.01 |
| 9 | 283.89 | 4.29 |  |  |  | 4.29 | Co3/Co4 | 4286 | 6.19 |
| 9 | 284.59 | 4.30 |  |  |  | 4.30 | Co3/Co4 | 4286 | 9.91 |
| 10 |  |  | 129.88 | 4.32 |  | 4.32 | CN1/CN2 | 4327 | −7.96 |
| 11 | 296.27 | 4.36 | 130.51 | 4.33 | −28.85 | 4.34 | CN3/CN4 | 4356 | −12.45 |
| 11 | 296.83 | 4.36 | 130.98 | 4.34 | −24.30 | 4.35 | CN3/CN4 | 4356 | −7.20 |
| 12 | 321.44 | 4.49 | 140.29 | 4.49 | −6.09 | 4.49 | N1/N2 | 4487 | 1.68 |
| 12 | 322.20 | 4.49 | 141.26 | 4.50 | 1.42 | 4.49 | N1/N2 | 4487 | 7.57 |
| 13 | 327.99 | 4.50 | 144.21 | 4.51 | 5.08 | 4.51 | N3/N4 | 4523 | −17.10 |
| 14 | 370.48 | 4.57 |  |  |  | 4.57 | N5/N6 | 4570 | 3.13 |
| 14 | 371.15 | 4.57 |  |  |  | 4.57 | N5/N6 | 4570 | 4.24 |
| 15 | 405.74 | 4.63 | 171.57 | 4.63 | −0.65 | 4.63 | N9/N10 | 4648 | −17.12 |
| 15 | 406.52 | 4.64 | 172.40 | 4.66 | 16.61 | 4.65 | N9/N10 | 4648 | −0.58 |
| 16 | 422.22 | 4.80 | 177.50 | 4.81 | 9.95 | 4.80 | Si1/Si2 | 4807 | −3.63 |
| 16 | 422.74 | 4.80 | 178.28 | 4.82 | 14.91 | 4.81 | Si1/Si2 | 4807 | 2.27 |
| 17 | 433.80 | 4.87 | 184.22 | 4.88 | 11.08 | 4.87 | Si5/Si6 | 4883 | −8.01 |
| 18 | 463.33 | 4.96 | 196.22 | 4.95 | −8.63 | 4.95 | ST3/ST4 | 4976 | −22.06 |
| 19 | 483.24 | 5.01 | 204.98 | 4.99 | −12.78 | 5.00 | T1/T2 | 5002 | −1.50 |
| 19 | 483.89 | 5.01 | 205.52 | 5.00 | −11.56 | 5.00 | T1/T2 | 5002 | 0.63 |
| 20 | 509.33 | 5.07 | 216.35 | 5.05 | −20.01 | 5.06 | T3/T4 | 5038 | 19.88 |
| 20 | 510.27 | 5.07 | 216.92 | 5.05 | −19.52 | 5.06 | T3/T4 | 5038 | 22.32 |
| 21 | 531.87 | 5.12 | 229.59 | 5.11 | −10.33 | 5.12 | T5/T6 | 5094 | 21.42 |
| 21 | 532.47 | 5.12 | 230.03 | 5.11 | −9.66 | 5.12 | T5/T6 | 5094 | 23.16 |
| 22 | 550.12 | 5.16 | 240.01 | 5.16 | −3.91 | 5.16 | T7/T8 | 5165 | −3.70 |
| 22 | 550.85 | 5.16 | 243.44 | 5.18 | 10.54 | 5.17 | T7/T8 | 5165 | 5.23 |
| 23 | 590.73 | 5.26 | 256.01 | 5.23 | −23.47 | 5.25 | TG1/TG2 | 5241 | 5.45 |

Glacier[52,54] is supported by a compilation of bulk geochemical data collected from outcrops on land[73] and compositions of detrital glaciomarine sediments that are also distinctly different between Pine Island Bay (in the eastern Amundsen Sea) and the central and western Amundsen Sea[54,55] (Fig. 1). For example, Zr/Hf ratios exhibit lower values in the Pine Island drainage basin than in Marie Byrd Land (Fig. 1), with other trace element ratios that are diagnostic for variations in petrology showing a similar spatial distribution (Supplementary Fig. 4). Even though data are sparse compared to the data set of Zr/Hf ratios, the compiled Eu/Eu* and $Gd_N/Yb_N$ values support the more mafic and younger, less evolved, magmatic signature of the West Antarctic Rift System/Marie Byrd Land province compared to the terrane underlying the Pine Island Glacier drainage basin.

IRD in the greenish grey interglacial units A12 and A13 (~4.5 − 4.4 Ma) includes grains with a very pronounced $^{40}Ar/^{39}Ar$ age peak of ~180 Ma, which points to iceberg sources in the Pine Island Glacier system (Fig. 8). Through application of $^{40}Ar/^{39}Ar$

geochronology to hornblende and biotite grains in recent glaciomarine sediments, Simões Pereira et al.[54,55] found $^{40}Ar/^{39}Ar$ ages between 250 and 170 Ma in the eastern Amundsen Sea Embayment offshore Pine Island Glacier. These ages overlap with ages of coarse-grained debris at the base of the Byrd Ice Core (~290 − 180 Ma), and represent contributions from the Jurassic−Triassic evolved granitoid source dominating $^{40}Ar/^{39}Ar$ ages of granitoids of the Thurston Island/Pine Island Glacier block and Ellsworth-Whitmore Mountains block[52,54,55]. Notably, this mode is poorly represented in glaciomarine sediments collected offshore from Thwaites Glacier[54,55]. In contrast, these sediments are mainly characterized by $^{40}Ar/^{39}Ar$ ages ranging between ~330 and 35 Ma, with a predominant ~115 − 110 Ma mode attributed to a mid-Cretaceous source under Thwaites Glacier, and a minor mode at ~330 − 280 Ma, coeval with magmatic activity recorded in Marie Byrd Land[54,55]. There are only few grains indicative of a Thwaites source, i.e. younger than 115 Ma or older than 250 Ma, in the 4.5 − 4.4 Ma interval (A12 − A13). The Eu anomaly values in the terrigenous muds deposited

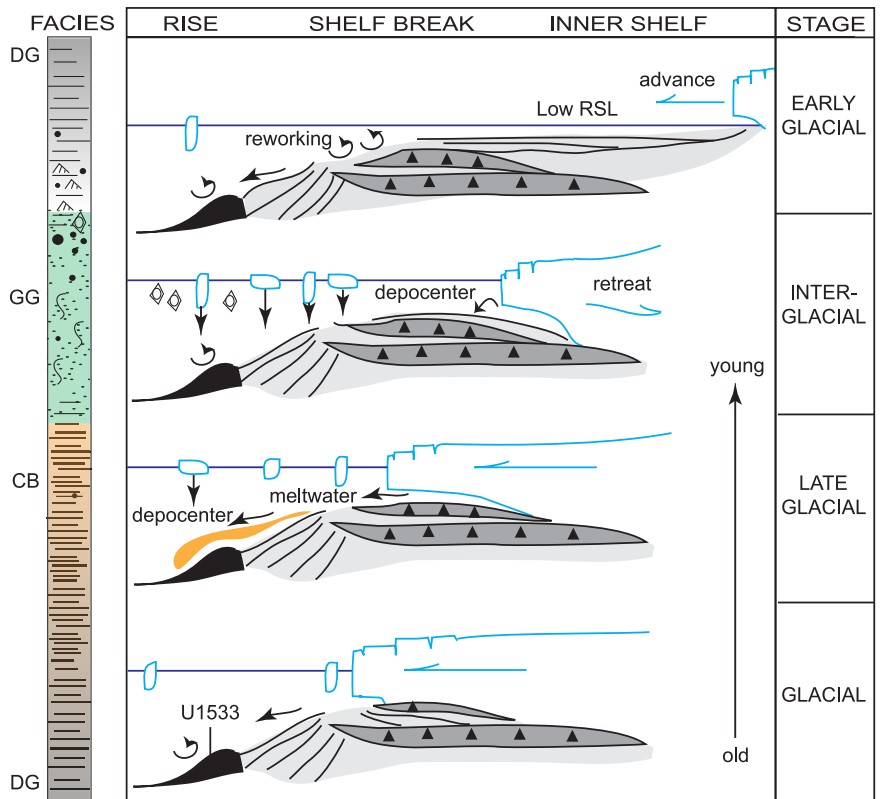

**Fig. 6 | Facies sequence and interpretation throughout one glacial cycle in the early Pliocene section of IODP Site U1533 on the Amundsen Sea continental rise.** DG dark grey laminated silty clay; CB color-banded silty clay; GG greenish grey silty clay; RSL relative sea level. Black triangles represent subglacial deposition of sediment. Source: modified from Passchier et al. (2019; their Fig. 10)[41].

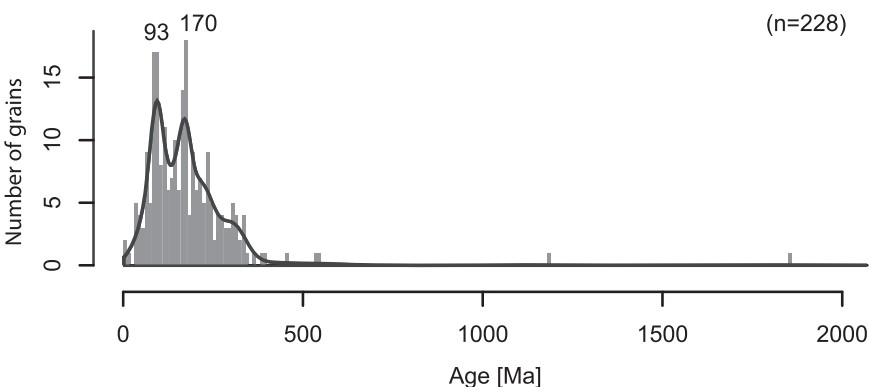

**Fig. 7 | Histogram and density distribution for ⁴⁰Ar/³⁹Ar ages of 228 individual ice-rafted ( > 250 μm) hornblende, biotite and mica grains derived from the early Pliocene greenish grey facies at sites U1532 and U1533.** Bin width is 10 Myr. Source data are provided as a Source Data file.

on the rise in this interval (Fig. 10) also point to a supply of detritus from Pine Island Glacier, which is characterized with Eu/Eu* values as low as 0.51 – 0.59 in the mud fraction of modern sediments[55]. Given that the Pine Island Glacier drainage system supplied detritus to the continental rise during glacials, it is not likely that the general lack of Thwaites Glacier material can be explained by trapping of sediment on the Amundsen Sea continental shelf.

The abrupt increase and gradual decrease in kaolinite and $Gd_N$/$Yb_N$ ratios to a minimum ~4.6 – 4.4 Ma may represent dynamic WAIS thinning followed by wide-spread and prolonged ungrounding of ice in the marine-based sectors of West Antarctica, most likely Thwaites Glacier as this is the largest source of kaolinite in the Amundsen Sea (Fig. 10). Pre-Oligocene sedimentary strata are a likely source of

modern ( >30%) kaolinite bearing sediments[54,55] and are detected under Thwaites Glacier in aerogeophysical surveys, cross-cutting the drainage conduit ~150 – 200 km landward of the grounding zone[48]. The declining kaolinite contents ~4.6 – 4.4 Ma, even in glacials, indicate prolonged retreat of the Thwaites Glacier grounding zone, to probably more than ~200 km landward from its present position and up-stream from any likely considerable kaolinite source. It is possible that the thick sediment drapes on the buried GZWs observed in the seismic profiles of the Amundsen Sea shelf result from such a period during which grounded ice resided in a retreated position for >200 kyr[37].

Spill-over of dense shelf water masses with high suspended sediment load originating near grounding zones[9] and hyperpycnal transport downslope is a likely mechanism for the deposition of the color-

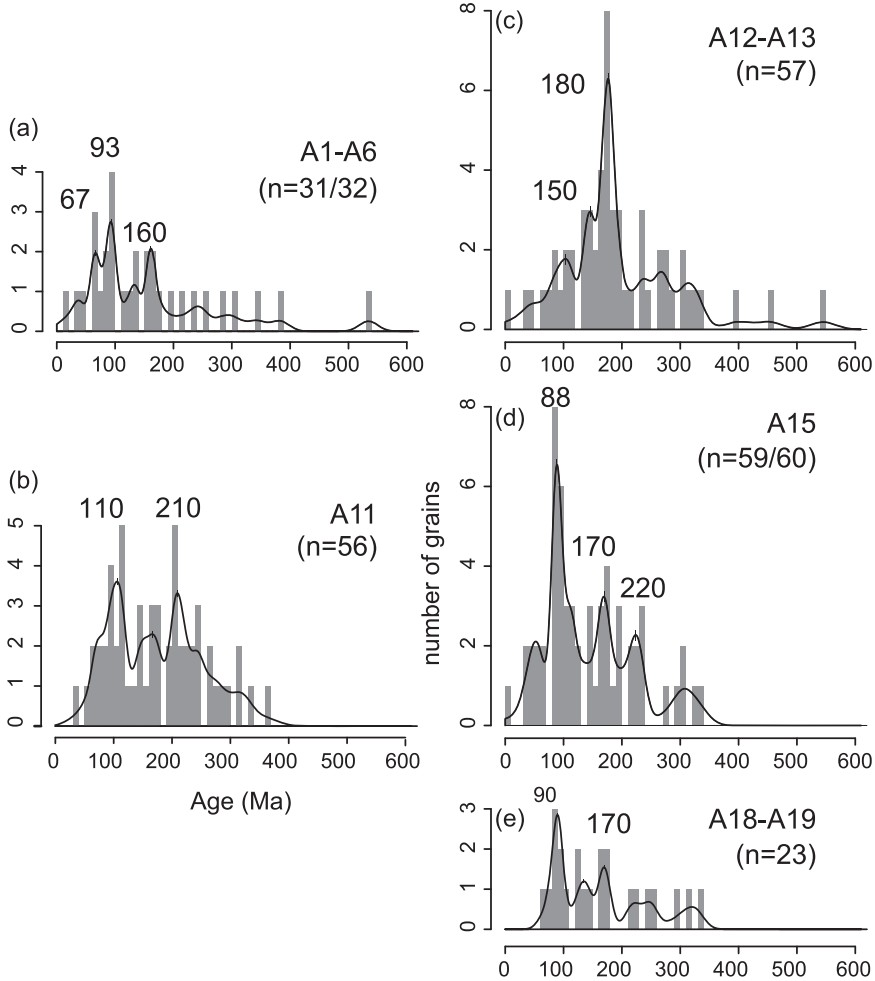

**Fig. 8 | Histogram and density distribution for $^{40}$Ar/$^{39}$Ar ages of ice-rafted ( >250 μm) hornblende, biotite and mica grains in different intervals.** The grains are derived from the early Pliocene greenish grey units (**a**) A1-A6, (**b**) A11, (**c**) A12-A13, (**d**) A15, and (**e**) A18-A19, at sites U1532 and U1533. Very few grains were dated as older than 600 Ma and are not shown here. Bin width is 10 Myr. An interval that generally lacks Cretaceous and Permian-Carboniferous $^{40}$Ar/$^{39}$Ar ages suggests an ice-rafted debris (IRD) source in the Pine Island Glacier drainage basin (IRD intervals A12-A13), whereas a maximum of IRD grains with an age of ~110 Ma and the presence of pre-Triassic IRD grains (IRD intervals A1-A6 and A11) points to a provenance in the Thwaites Glacier basin. Source data are provided as a Source Data file.

banded facies (Fig. 6), which accumulated at very high linear sedimentation rates (up to ~900 m/Myr) at ~4.6 – 4.5 Ma (Fig. 10). Hyperpycnal plumes and flows often originate from processes that enhance the formation of high-density shelf waters through changes in suspended sediment load, salinity and temperature, such as late glacial river or subglacial meltwater discharge[74–77], brine rejection during sea-ice formation[78] and/or supercooling in ice-shelf cavities[79]. Relatively small and invariant mean sortable silt grain sizes in all facies (Fig. 10) support a weak bottom current regime on the continental rise, much like sediment records from the sub-Antarctic Pacific sector show a weak ACC circulation during this early Pliocene interval[72]. Weak bottom currents allowed the rapid deposition of mud from meltwater plumes originating in the Amundsen Sea that extended to abyssal water depths more than 250 km from the shelf break.

Increased kaolinite contents ( >12%) at 4.3 – 4.1 Ma (Fig. 10) point to increased supply of glaciogenic debris from Thwaites Glacier following the ice-sheet minimum, because this system is the strongest kaolinite source in the Amundsen Sea Embayment today. Correlative $Gd_N/Yb_N$ and kaolinite content increases support drainage of ice originating from the Thwaites Glacier system at this time[54,55,58]. Grains of early Cretaceous ( ~110 Ma) and older ages (up to ~400 Ma) in the IRD-rich bed ~4.3 Ma also suggest renewed supply from Thwaites Glacier (A11 in Fig. 8). This provenance signature can be interpreted as re-grounding or renewed ice advance in the Thwaites Glacier catchment

by ~4.3 Ma, with glacial-interglacial oscillations of the grounding zone. WAIS growth during that time was likely augmented by subsurface ocean cooling, reflected by the decrease in SST at DSDP Site 590[27] (Fig. 10), and the expected glacio-isostatic uplift in response to prior dynamic thinning of the WAIS[80].

After ~4.1 Ma, a considerable decrease or cessation of dense shelf water mass production and/or trapping of glacial meltwater plumes on the Amundsen Sea continental shelf likely reduced the supply of fine-grained suspended detritus by hyperpycnal flows to the continental slope and rise, similar to the modern situation. An increase in biogenic silica content of the sediments at both sites[39,40] (Fig. 3 and Supplementary Fig. 3) can be attributed to a combination of less dilution by terrigenous supply and increased diatom preservation at shallower depth in the formation. Linear sedimentation rates on the continental rise decreased further (Supplementary Fig. 5), indicating an overall reduction in terrigenous supply to the continental rise. Mean sortable silt grain size and smectite contents correlate positively and reach maxima during interglacials, indicating a terrigenous component being reworked by more active bottom currents. The provenance data for the ~4.1 – 3.6 Ma Amundsen Sea sediments indicate WAIS stability during glacials and interglacials with a generally more expanded ice sheet. Sparse IRD with more grains of Cretaceous age in interglacials (A1 – A6) indicates a return to "background" low-intensity ice rafting ~4.1 Ma, whereas increased $Gd_N/Yb_N$ and Eu/Eu* ratios in the glacial-

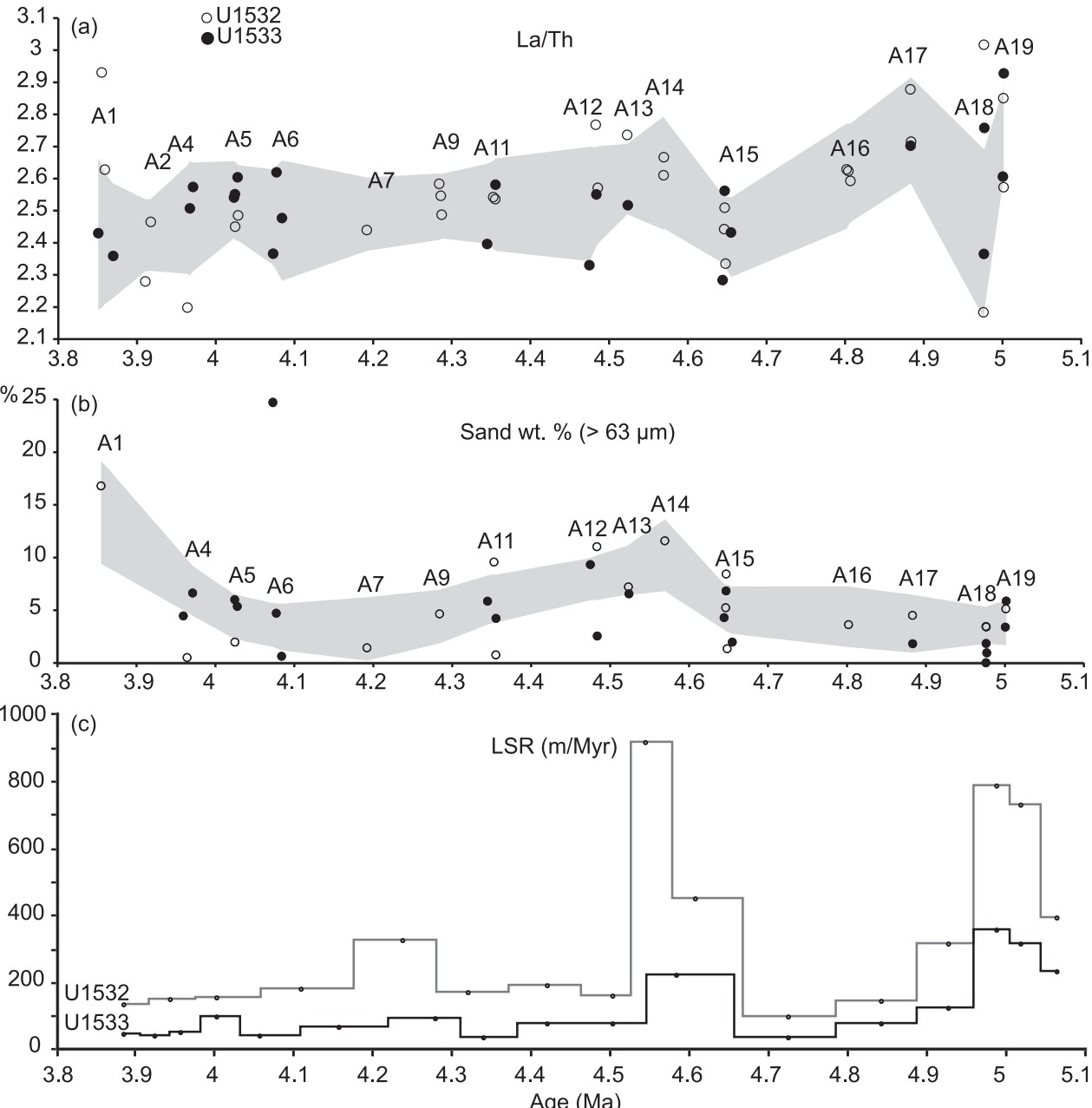

**Fig. 9 | Geochemical provenance variability in correlative greenish grey interglacial beds with IRD at Sites U1532 and U1533.** Grey shading designates a 97.5 % confidence envelope of a LOWESS regression model fitted through the data. **a** La/Th values are highly variable in ice-rafted debris (IRD)-bearing interglacial beds deposited between 5.1 and 4.5 Ma, and relatively constant and low in IRD-bearing interglacial beds deposited between ~4.5 and 3.9 Ma. Higher La/Th values indicate an arc-volcanic source in the Cretaceous-Cenozoic central and western domain of West Antarctica and/or a larger pelagic component, whereas lower La/Th ratios coincide with intervals with a larger terrigenous component and a source within the Triassic and older terrains of the eastern domain (Fig. 1 and Supplementary Fig. 1). **b** Sand wt.% ( >63 µm) in samples from the greenish grey facies are high between ~4.6 and 4.5 Ma. **c** Linear sedimentation rates (LSR in m/Myr) are high around ~5 Ma and between ~4.6 and 4.5 Ma. The age model and LSR are based on paleomagnetic age tie points and correlations of greenish grey facies to the ages of glacial termi-nations in the LR04 benthic stack[69,95]. Source data are provided as a Source Data file.

stage terrigenous silts deposited after this time (Fig. 3) are consistent with the delivery of glacigenic detritus from an additional source of sediments in the younger arc volcanic terranes of the western domain of West Antarctica[55]. The interglacial increases in bottom current speed recorded ~4.1 – 3.6 Ma on the Amundsen Sea continental rise coincide with an intensification in ACC flow[72], whereas SSTs at DSDP Site 590, indicative of Antarctic subsurface ocean temperatures, show

an overall cooling trend[27] (Fig. 10). These correlations confirm the strong coupling between ice-sheet extent and Southern Ocean circulation on timescales longer than the observational record as previously proposed in Pliocene modeling experiments[70].

Counterintuitively, ice cover and strong bottom currents at Sites U1532 and U1533 in the earliest part of the Pliocene coincide with sustained ocean warming over glacial-interglacial cycles (Fig. 10). One

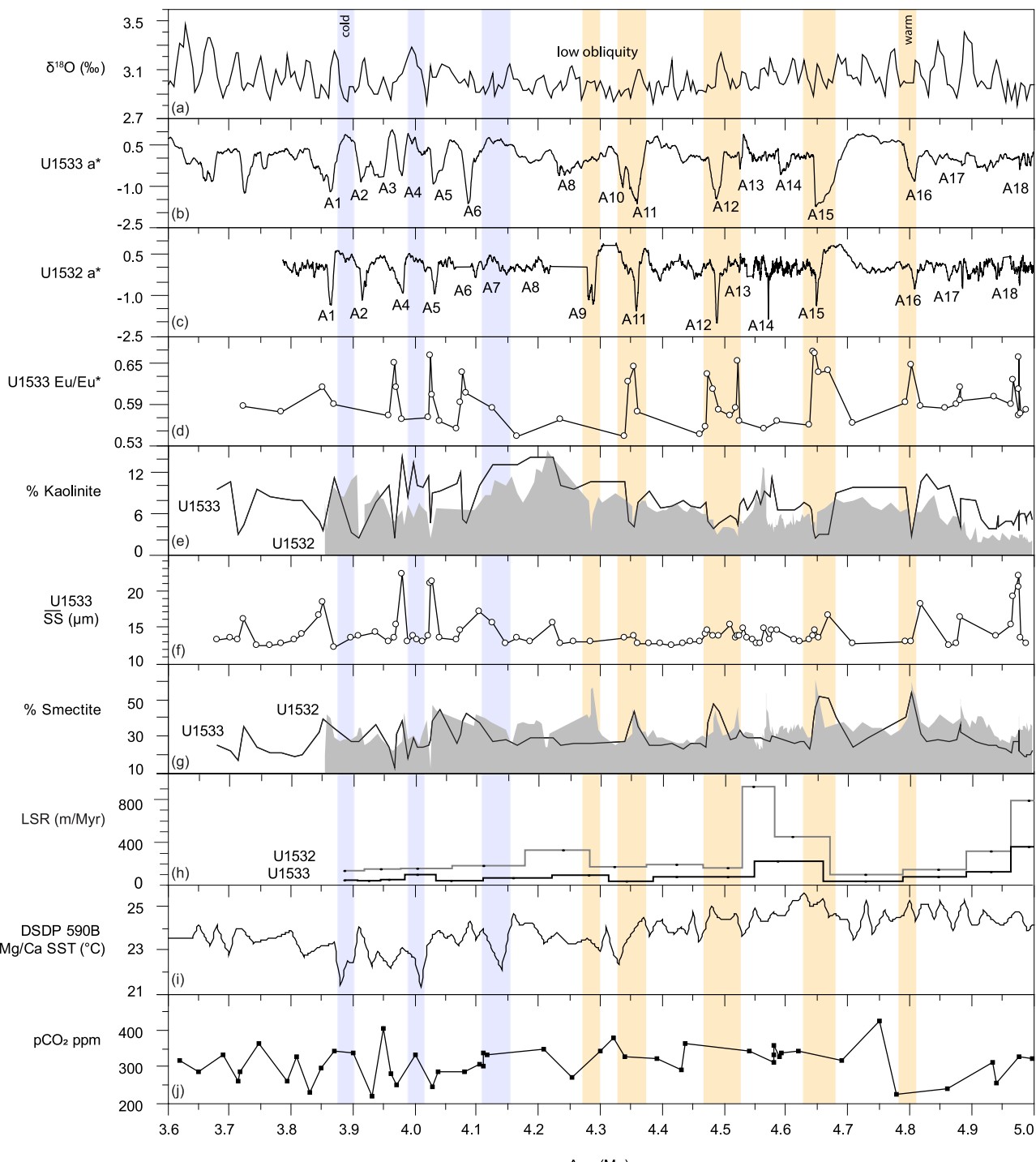

**Fig. 10 | Time series of Site U1532 and U1533 datasets compared to sea-surface temperature (SST) and pCO₂ proxies.** Panel (**a**) LR04 benthic stack[69,95]; **b** a* color photospectrometry record of Site U1533, with low values indicating green and high values red hues; **c** a* color photospectrometry record of Site U1532; **d** Europium anomaly provenance indicator calculated from the bulk geochemistry of U1533 samples; **e** Kaolinite % in the <2 μm size fraction, with the black line connecting the data points for Site U1532 and the outline of the grey shading the data for Site U1533; **f** Mean sortable silt grain size ($\overline{SS}$) current strength proxy; **g** Smectite % in the <2 μm size fraction, with the black line connecting the data points for Site U1532 and the outline of the grey shading the data for Site U1533; **h** linear sedimentation rates (LSR in m/Myr); **i** Foraminiferal Ca/Mg based SST record for DSDP Site 590 from Karas et al. [27]; **j** pCO₂ reconstructions based on a compilation of the CenCO₂PIP Consortium[100], using boron and phytoplankton proxy data[26,101–103]. The age model is based on paleomagnetic age tie points and correlations of greenish grey facies to the ages of glacial terminations in the LR04 benthic stack[69,95]. Source data are provided as a Source Data file.

explanation for phases of ice advance during peak ocean warming could be enhanced meridional moisture transport and accumulation, which is strongly influenced by obliquity. The glacial-interglacial cyclicity as expressed in the a* records appears to be characterized by an obliquity beat before 4.5 Ma, in conjunction with strong obliquity forcing of the cryosphere[69,81] (Fig. 10). In contrast, between 4.5 and 4.1 Ma, fewer complete sequences with greenish grey facies with Ba/Rb maxima are present in the Amundsen Sea continental rise stratigraphy than there are interglacials in the LR04 benthic stack, through a period when the glacial-interglacial amplitude is also muted (Fig. 10).

Suspended sediment supply during interglacials may have altered the visual or compositional signature of the hemipelagic sedimentation by diluting authigenic and biogenic sediment components.

Ice loss from the major Amundsen Sea sector drainage basin commenced ~4.6 – 4.5 Ma under prominent but weakening obliquity forcing[81] during a period with high SSTs in the South Pacific[27], and high levels of pCO$_2$, conditions that are similar to the Earth System's current trajectory. In a more temperate glacial setting the development of fast flowing ice streams at the termination of glacial phases increased ice-marginal erosion rates and offshore supply of glaciogenic debris, even when grounded ice did not reach the shelf break[62]. Under prolonged higher surface temperatures in the early Pliocene, the relatively low elevation of the ice sheet augmented by ocean-driven dynamic thinning could have made WAIS also more susceptible to relatively abrupt atmospheric warming and surface melt[17]. The production of hyperpycnal flows, with increased suspended sediment volume from ice melt, indicates the ungrounding of parts of the WAIS in its Amundsen Sea sector that likely contributed to the ~20 m early Pliocene sea-level highstand ~4.4 Ma[28]. Meltwater export and grounded ice loss also likely affected Southern Ocean circulation as previously proposed in modeling experiments[70]. The Amundsen Sea Embayment is currently not a region of deep-water formation. However, under conditions of sustained warming and retreat in the early Pliocene, spill-over of dense shelf water off West Antarctica likely changed density gradients in high-latitude deep-water masses, with potential consequences for oceanic heat transport and CO$_2$ sequestration.

## Methods

Shipboard, lithological descriptions were carried out on the archive halves of the cores based mainly on visual characteristics, including color, grain-size and sedimentary structures[82]. Cores were imaged using a Section Half Image Logger (SHIL). Biogenic components were identified and estimated using smear slides. Archive halves were also analyzed with a Section Half Multisensor Logger (SHMSL) at 2 cm intervals. The sampling strategy was designed to avoid aliasing and shipboard track data, in particular a* color reflectance and magnetic susceptibility records, were used to identify different phases of interpreted interglacial–glacial cycles and to sample all representative facies. Subbottom depths were originally reported as m CSF-A[82], and these depths are re-designated in this study as meters below seafloor (mbsf) for clarity. For this study, sediments from Hole U1533B between ~84 and 205 mbsf and Holes U1532C and U1532G between ~197 and 531 mbsf, dated between ~3.6 and 5.0 Ma, were analyzed. The shipboard age model was refined through correlation of interglacial beds of greenish grey facies between the two drillsites U1532 and U1533 and tied to orbitally tuned excursions in the LR04 benthic stack[69].

### Grain-size analysis

Particle size data was collected on 94 samples from Site U1533 to characterize the grain-size variability both within each facies and between the facies. The size fraction 0.02 to 2000 μm of Site U1533 sediments was measured with a Malvern Mastersizer 2000 laser particle sizer at Montclair State University (U.S.A.). This is a dual light source instrument with a large array of detectors and software capabilities including Mie theory for the accurate and precise characterization of fine-grained sediments. Sample pretreatment consisted of boiling the sediments with 30% H$_2$O$_2$ and 10% HCl to remove organic matter and carbonate. Where diatoms were present, samples were heated with 0.2 NaOH in a hot bath at 90 °C for one hour to remove biogenic silica. Sodium pyrophosphate was used as a dispersant. Samples were measured using a Hydro MU sample dispersion unit with a stirred speed of 2000 rotations per minute. Grain-size distributions were generated using the Malvern software using a refractive index of 1.6 and an absorption coefficient of 0.9. Sortable silt mean size was calculated from the bulk particle size distributions. Sortable silt refers

to the non-cohesive fine-sediment fraction with a size range from 10 to 63 μm. Sortable silt percent (SS%) and mean sortable silt grain size ($\overline{SS}$) were calculated using the equations[45,83]:

$$SS\% = vol.\%[10 - 63\mu m]/vol.\%[< 63\mu m]*100 \tag{1}$$

$$\overline{SS} = e^{\wedge}\theta_m \ with \ \theta_m = \sum(i=1,n)(\theta_i)fss,i \tag{2}$$

($\theta$i) is the natural log of the bin's geometric mid-point diameters and fss,i is the fraction of material associated with the SS range in bin i, with bin 1 representing the bin with its lower bound coinciding with 10 μm, and bin n representing the bin with its upper bound coinciding with 63 μm.

Positive correlations between sortable silt percent and mean sortable silt grain size are interpreted to represent sorting by bottom currents, and this holds for most sediments even when influenced by particle deposition from icebergs, sea ice or meltwater plumes. High IRD input can mask the bottom-current signal, and this is revealed by a poor correlation between the percentage and mean grain size of the sortable silt[45].

### Clay mineralogy

Clay mineral assemblages were analyzed at the University of Leipzig (Germany) following standard procedures[58]. The clay fraction ( <2 μm) was separated from the bulk sediment in settling tubes. 40 mg clay were dispersed in an ultrasonic bath and sucked through a membrane filter of 0.20 μm pore width. The dried filter cakes were mounted on aluminium tiles and glycolated immediately before the X-ray analyses. Samples were X-rayed on a Rigaku MiniFlex automated powder diffractometer system with CoKα radiation (30 kV, 15 mA) in the range of 3°–40°2θ with a step size of 0.02°2θ and a measuring time of 2 s/step, and in the range of 27.5°–30.6°2θ with a step size of 0.01°2θ and a measuring time of 4 s/step to better resolve the (002) kaolinite and (004) chlorite peaks. The X-ray diffractograms were evaluated using the interactive "MacDiff" program (freeware available from https://www.uni-frankfurt.de/69528130/Petschick__MacOS_Software). The semiquantitative evaluation of the clay mineral assemblages followed Biscaye[84,85].

### Bulk geochemical composition

Post-cruise, archive halves of Hole U1533B were scanned at 2 cm resolution with an energy of 10 kV, 30 kV, and 50 kV on an Avaatech XRF core scanner at the IODP Gulf Coast Repository of Texas A&M University in College Station (U.S.A.). in order to provide a record of elemental values for the whole sequence (84 – 205 mbsf) Specifically, XRF results for Ca, K, Ti, Mn, Rb, Zr, Nb, Ba were used to investigate elemental ratios (Ca/Ti, K/Ti, Mn/Ti, Zr/Rb, Ba/Rb, and Ti/Nb) indicative of biogenic and detrital sediment supply and composition. Bulk geochemical data were collected on 108 samples from sites U1532 and U1533 using a ThermoFisher Quadrupole ICP-MS at Montclair State University (U.S.A.). Approximately 100 mg of powdered sample and 400 mg of lithium metaborate flux were quenched at 1050 °C and digested in 7% nitric acid. The solution was diluted to 0.1% and sample unknowns were analyzed in triplicates with 10 USGS calibration standards. Measurements of standard solutions of Ba, Ce, Pr, Nd and Sm were applied for interference corrections. Ratios of trace and Rare Earth Elements (REE) were interpreted to assess mineral content and bulk provenance for the mudstones. The cerium anomaly (Ce/Ce*) was utilized as a proxy for authigenic mineral content in which the deviation of Ce concentrations is calculated with respect to a smooth REE pattern, with 'SN' designating shale-normalized values[86]:

$$Ce/Ce^* = Ce_{SN}/((La_{SN} + Pr_{SN})/2) \tag{3}$$

## Provenance from elemental ratios

Ratios of compatible to incompatible elements were used to infer geochemical provenance. During magmatic processes, compatible elements are incorporated into minerals whereas incompatible elements remain in the melt. We use non-metric multidimensional scaling with Bray-Curtis similarity of a multivariate dataset of elemental ratios to identify similarities in provenance between source rocks and Expedition 379 sediments[87]. The statistical analyses were carried out in the software package PAST v. 4.09[88]. The elemental ratios $Gd_N/Yb_N$, La/Th, Sm/Zr, Th/Sc, Zr/Hf and Eu/Eu* differentiate between types of crustal rocks[89,90] and show clear spatial distributions in existing source rock and sediment data in West Antarctica[54,55,73,91] (Fig. 1; Supplementary Fig. 4). 'N' stands for chondrite-normalized values according to Bhatia[86]. $Gd_N/Yb_N$ ratios track the relative abundance of heavy rare earth elements (HREE) and can be used to distinguish different felsic igneous source rocks, ranging from K-rich granites with stable HREE patterns and low $Gd_N/Yb_N$ to more Na-rich granodiorite with higher $Gd_N/Yb_N$ ratios, which represent different processes of crustal formation[92]. Th/Sc ratios are low and La/Th ratios are high in mafic to intermediate igneous rocks[89,90] and La/Th ratios differentiate arc-volcanic sources due to inheritance of these ratios from the subducting crust[72]. Magmatic source rocks can further be distinguished using Sm/Zr and Zr/Hf ratios, which are hosted in the chemically and mechanically resistant accessory minerals of monazite, zircon and baddeleyite. Monazite is enriched in Sm and more easily released upon partial melt[93], whereas Zr is more easily taken up than Hf upon crystallization of zircon, allowing for compositional differentiation of volcanic and plutonic igneous source rocks[94]. Europium is a compatible element and partitions into plagioclase upon crystallization in magma chambers, so residual magmas display a negative europium anomaly (high value of Eu/Eu*). The europium anomaly is expressed as Eu/Eu*, where Eu* is the anticipated europium value for a smooth chondrite normalized REE pattern versus atomic number[86].

$$Eu/Eu^* = Eu_N/((Sm_N + Gd_N)/2) \qquad (4)$$

Different calculations of the europium anomaly are used by individual authors and where sample Eu/Eu* values are compared to existing published data, the Eu/Eu* ratios for the published data are recalculated using the same equation from Bathia[86].

## $^{40}Ar/^{39}Ar$ geochronology of coarse sand fraction

$^{40}Ar/^{39}Ar$ geochronology was applied to 230 hornblende, biotite and other mica grains picked from the sieved >250 μm fractions of samples from the greenish grey units, interpreted as coarse-grained IRD. $^{40}Ar/^{39}Ar$ determinations were performed at the Argon Geochronology for the Earth Sciences (AGES) Laboratory at Lamont Doherty Earth Observatory of Columbia University (U.S.A.), using a fully automated Micromass VG 5400 mass spectrometer with a $CO_2$ laser extraction system and operated by MassSpec software from Al Deino (Berkeley Geochronology Center). Minerals were co-irradiated with monitor standards offsite to produce $^{39}Ar$ from $^{39}K$ and to define the "J-value", used for the age calculations.

## Age model

We used the shipboard age models for Sites U1532 and U1533[39,40], which in the Pliocene are based on magnetostratigraphic age tie points constrained with diatom and radiolarian biostratigraphy (Supplementary Fig. 5a). We refined the shipboard age models by correlating marker beds between the two drillsites and then further refined the age model through correlation of the sequence stratigraphy to the LR04 benthic stack[69,95] (Supplementary Figs. 5b and 5c).

## Data availability

Shipboard data and observations are hosted on Zenodo at https://zenodo.org/communities/iodp/records and an overview of available data can be found at the International Ocean Discovery Program http://publications.iodp.org/proceedings/379/datasets.html. The XRF core scanner data for Site U1533 generated in this study have been deposited in the database of the Joides Resolution Science Operator (JRSO) under accession code: https://doi.org/10.5281/zenodo.11200517. Bulk geochemical major, trace and rare earth elemental data on discrete samples of Sites U1532 and U1533 generated in this study are available in the EarthChem data repository under accession codes: https://doi.org/10.60520/IEDA/113370 https://doi.org/10.60520/IEDA/113371. Clay mineral data for Sites U1532 and U1533 generated in this study are available from the Polar Data Centre at the British Antarctic Survey under accession codes: http://ramadda.data.bas.ac.uk/repository/entry/show?entryid=41524ab6-e19d-43d9-8ba1-8ff2fab1a955, https://ramadda.data.bas.ac.uk/repository/entry/show?entryid=f3c8c5e2-f8c2-4938-a7e0-6fdc9bedb412 and https://ramadda.data.bas.ac.uk/repository/entry/show?entryid=3b855b7b-fb14-439a-a6d9-ac233133cf23. Grain size data for Site U1533 generated in this study have been deposited in the US Antarctic Program database under accession code: https://doi.org/10.15784/601900. $^{40}Ar/^{39}Ar$ data for >250 μm coarse fractions from Sites U1532 and U1533 generated in this study have been deposited in the US Antarctic Program database under accession code: https://doi.org/10.15784/601907 Source data are provided with this paper.

## Code availability

All software packages utilized in this study are publicly accessible, and no original code is reported in this study. Histograms and density distributions for $^{40}Ar/^{39}Ar$ ages in Figs. 7 and 8 were generated using the online version of IsoplotR[96], available at this link: https://www.ucl.ac.uk/~ucfbpve/isoplotr/home/ For the statistical analyses (MDS Fig. 5 and Loess smoothing in Fig. 9) the software package PAST v. 4.09[88] was used, that can be downloaded from this link: https://www.nhm.uio.no/english/research/resources/past/index.html

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

## Acknowledgements

This work used data and samples provided by the International Ocean Discovery Program. Expedition 379 Scientists are thanked for shipboard work. Josie Horowitz collected XRF data at Texas A&M University in College Station (funded by Montclair State University). Sylvia Haeßner carried out laboratory work at the University of Leipzig and Jessica Scheinbaum and Xiaona Li are thanked for sample prep and ICP-MS lab assistance at Montclair State University. Ronald Leon, Lisbeth Mino-Moreira and Jessica Scheinbaum were supported by the U.S. National Science Foundation through the Garden State-LSAMP (NSF Award 1909824). Additional funding for this work was provided by the U.S. Science Support Program of the IODP, the U.S. National Science Foundation (NSF Award 2114839 to SP) and the UK's Natural Environment Research Council (UK-IODP grant NE/T010975/1 to CDH).

## Author contributions

Conceptualization: S.P., C.D.H. and T.F.; Data curation: S.M.B., S.R.H., C.D.H., S.P.; Formal Analysis: S.P., C.D.H., W.E., S.R.H.,T.F.; Funding acquisition: S.P., C.D.H., J.W.; Investigation: S.M.B., S.R.H., R.L., O.L.R., L.M.M., S.P.; Project administration: S.P., Resources: K.G., J.W.; Writing – original draft: S.P.; Writing – review & editing: C.D.H., W.E., T.F., K.G., J.W., S.M.B., O.L.R.

## Competing interests

The authors declare no competing interests.
