## [Peer Review file · Nature Communications]

West Antarctic ice retreat and paleoceanography in the Amundsen Sea in the warm early Pliocene

Corresponding Author: Dr Sandra Passchier

Parts of this Peer Review File have been redacted as indicated to remove third-party material where no permission to publish could be obtained.

Version 0:

Reviewer comments:

Reviewer #1

(Remarks to the Author)

The manuscript from Passchier et al. is well written and addresses an important topic that has interest for the larger Earth science community and beyond.

Passchier et al. applies a multi proxy approach to infer sediment provenance and processes from a really nice sediment stratigraphy from the Amundsen Sea continental rise.

Based on the data they present and their discussion, I find their conclusions that grounded ice in the Twaites sector of the West Antarctic Ice Sheet retreated during the early Pliocene both robust and compelling.

While the figures and text are generally clearly presented, I feel that the manuscript would have benefitted from illustrations of the processes behind the different sedimentary facies encountered in the record, including the modern processes that is referenced several times in the manuscript but not thoroughly described. This would especially benefit non-specialists reading this paper.

Some comments on figures: The figures is generally clearly presented and readable, however I miss some things regarding captions and labeling etc:

Fig. 4: Units of the a* and mSS panels are missing (I suppose the mSS should be μm)

Fig. 5: the a* records lacks scale (where are the low values..)

Fig. 8: lowermost panel is not mentioned in the caption, it would also be useful to label the panels a)-C)

Fig. 9: labels a)-h) are missing in the figure

Reviewer #2

(Remarks to the Author)

The manuscript investigates cored materials from the Amundsen Sea in the Pliocene to explore the coeval evolution of the West Antarctic Ice Sheet and oceanographic conditions. The methods presented are robust (though some of the geochemical methods I confess to only having a working understanding of rather than a detailed knowledge) and the inferences from the results are fair and well argued.

In reality, with the caveat of my gap in knowledge of the geochemical methods, this was a very easy paper to review and I had very little to negatively comment upon in the manuscript. It appears to be in a very mature condition, it is well written, the figures help to make the case, and throughout the context is well referenced. Given the potential analogy of this warm period the authors investigate, I would state that I think the paper provides an important contribution - though the authors could perhaps consider elaborating on this further. I have some minor suggestions in the pdf.

Reviewer #3

(Remarks to the Author)

The authors present an interesting manuscript aimed at understanding the glacial history of the West Antarctic Ice Sheet (WAIS) by examining sediment archives from the deep-sea Antarctic Continental Rise, located downstream from the "doomsday" glaciers of Pine Island and Thwaites. Their goal is to use various proxies (color, grain size, XRF, $^{40}\text{Ar}/^{39}\text{Ar}$) to interpret the extent and dynamics of different WAIS glaciers in relation to Pliocene climate and oceanography.

The study design is solid, the methods align well with the objectives, and I did not find major flaws in the data interpretation. However, after multiple readings, I believe several significant issues need to be addressed before the manuscript can be considered for publication.

Below, I provide major comments to help the authors enhance their manuscript, followed by specific feedback in the marked-up version.

Major comment 1:

The manuscript lacks a clearly articulated research question or problem, which is reflected in the overall structure (see also Major Comments 2 and 3). The general aim appears to be an overview of WAIS evolution, primarily focusing on Pine Island and Thwaites glaciers during the Pliocene, to provide analogs of past warm periods and help constrain sea-level rise projections. However, sea-level only appears in the introduction and conclusion, and it's unclear whether the observed retreat from 4.6 to 4.4 Ma was more or less significant than today's retreat. The manuscript would benefit from a stronger comparison with present-day conditions to help readers grasp the research implications. A clearer focus on the broader problem, followed by the knowledge gap this study aims to address, would enhance the paper's impact. Additionally, the introduction could be better structured: start with the broader problem, state the knowledge gap, and present the methodology, followed by the regional setting and previous work. Much of the material currently in the methods and results sections, such as data from the IODP report and bedrock geology, belongs in this section. Also, geochemical provenance is barely mentioned in the conclusions, despite its focus in the methods.

Major comment 2:

The manuscript would greatly benefit from a more detailed results section. It is crucial to differentiate between what this study accomplished and what was published during IODP Expedition 379. The authors rely on previously interpreted lithofacies and use them as templates for sedimentation variations but present minimal new results. There are very few numbers or statistics provided, which makes it difficult to understand the dataset or assess the authors' interpretations. I recommend presenting the results from each analysis, such as mean element content by lithofacies. Currently, it feels like the results were skipped, and the manuscript jumps straight into a lengthy discussion.

Major comment 3:

The discussion would benefit first from an overview of the chronology/the sequence. As of now it is hard to follow track of what happened in chronological order.

You can start by a sub-section of discussion that states what that based on your results, the lithofacies 1 (terrigenous) exhibit characteristics a, b, c. Characteristic a can be associated with , characteristic b, can be associated with...etc. The authors kind of do a very brief version of that in the section "Interpretation of sedimentary facies", but I think it is too short – from the text it mostly/solely is: terrigenous facies is interpretation as glacial conditions based on the fact that it is terrigenous; color-banded is associated with the deglacial phase and ice-stream activation based on the high rates of sedimentation; and, the greenish facies are interpreted as interglacial conditions based on low Al/Ti ratios.

After you have interpreted the facies in term of what they mean in general conditions, present the sequence conditions : based on these facies you have 19 changes (A series) in background conditions that can be tied to LRD04 benthic stack. After correlating tie-points you have fewer A series than glacial-interglacial cycles as suggested by benthic stacks. Discuss why, and if it means something about the WAIS sensibility to orbital forcing + here you can introduce the variability of in the different A series in chronological order (IRD provenance, sedimentation rates, terrigenous sand content, kaolinite content, etc.) This will set the framework for the subsequent interpretations of changes in the cores.

Minor comments:

I have provided a marked-up Word version of the manuscript with detailed comments to make the revision process easier for the authors. This includes direct suggestions and queries.

Comments on figures:

Figure 1:

This figure is difficult to read due to the numerous black lines. Consider dividing it into two panels: one showing topography/bathymetry (with color), glacier extent, and catchments; and another showing geology and Zr/Hf values. Ensure that the colors in the legend match the map for clarity. Something like this with catchments on them and the age of each geological unit:

Figure 2:

The figure would be more effective if it showed a transition from glacial to interglacial (dark gray to color-banded to green), rather than starting with color-banded. Also, arrange the cores and history from oldest to youngest, left to right, to match the natural reading flow. Aligning all cores horizontally would make it easier to add curves and improve readability.

Figure 3:

Add transparency and outlines to the facies circles to ensure no data points are hidden by others.

Figure 4:

Include both cores and all curves referenced in the text. Consider removing the core number and recovery information, as these can be found in the IODP report. Align the lithology box with the analyzed interval and ensure the lithology reflects the facies discussed in the text. An idea would be to add the lithology color scheme to the whole figure/under the curves. This would really help with linking proxies and facies.

Figure 5:

Align the start and end points of the three curves and orient the figure horizontally.

Figure 6:

Add all argon-argon histograms from Figure 7. Remove the outline from the bins and use color fill instead to reduce visual clutter.

Figure 7:

If the curves are added to Figure 4, this figure may no longer be necessary.

Figure 8:

Label each chart with a, b, c, and explain each in the main text.

It's unclear where the data points originate. To improve clarity, I recommend differentiating between each core and deriving separate curves for each. This would help determine whether distinct trends or narratives emerge from the individual cores, which isn't evident in the current presentation.

Figure 9:

Excellent figure overall. However, I suggest flipping it to read from left to right, with the oldest data on the left. The gray area of core 32 is a bit lost and could just be in another color. You could present all curves that relates to core 33 in black and to core 32 in blue, in this figure and in all others. I would also add benthic stack at the top or bottom, it is the reference of the study.

Version 1:

Reviewer comments:

Reviewer #1

(Remarks to the Author)

Dear Passchier et al.,

I am satisfied with the reply to the reviewers comments and the changes made to the manuscript.
I recommend publication.

Reviewer #3

(Remarks to the Author)

I appreciate the efforts the authors have made in revising the manuscript. Overall, I find that the adjustments to the structure and the incorporation of suggestions have improved the readability and clarity of the paper. The manuscript is now more cohesive and better organized.

That said, I believe the introduction remains too long (currently spanning five pages) and contains some redundancies, particularly in the statement of the study's objectives, which are repeated multiple times (see marked-up copy). Additionally, I suggest that the last paragraph of the introduction be moved to the discussion, conclusions, or abstract, as it reveals key findings that should not be preemptively disclosed.

At this stage, my primary concerns are editorial in nature, and I have provided detailed suggestions in the marked-up copy. While I appreciate the inclusion of the new figure illustrating processes, I find that the revisions to the other figures were somewhat minimal. Rather than fully engaging with the constructive feedback, the authors have justified retaining their original figures and made only minor modifications. While these choices do not invalidate the study's conclusions, I believe they represent missed opportunities to enhance clarity and effectively communicate key results. Per example, I suggested that the core proxies be presented from left to right, progressing from older to younger, as this follows the natural reading order in English and aligns with how information is typically processed in an English-language journal. Since the authors are presenting values in years before present (i.e., negative values), the use of a standard x-axis convention—where zero is positioned on the right—would be more intuitive. Adjusting these aspects would improve clarity and better align with standard conventions for chronological data presentation.

Figures should prioritize the most essential information to support the paper's arguments. Currently, many figures feel overcrowded or could be optimized for readability. Below, I highlight some specific concerns:

Figure 1: The inclusion of hillshade beneath the ice introduces visual noise, making it difficult to discern key features.

Consider whether this layer is necessary. Additionally, the figure contains many overlapping black lines, reducing readability. A possible solution would be to split the figure into two panels: one focusing on topography/bathymetry (with glacier extent and catchments) and another dedicated to geology and Zr/Hf values. Ensuring that legend colors match the map would also improve clarity.

Figure 3: The resolution of the curves appears low or blurry. Additionally, the alignment of lithology, magnetic susceptibility, and core recovery should match the start of the curves for consistency.

Figure 5: The start and end points of the three curves should be aligned. It is unclear why the benthic stack appears to "float" on the left.

Figures 7 and 8: The current bin outlines add visual clutter. I recommend either removing the outlines and using color fills instead or add color to the curves to enhance contrast and improve readability.

While these are not fundamental issues, I believe addressing them would substantially improve the manuscript's presentation and accessibility to readers. I encourage the authors to consider these refinements to enhance the overall clarity of their figures.

REVIEWER COMMENTS

Responses in red.

Reviewer #1 (Remarks to the Author):

The manuscript from Passchier et al. is well written and addresses an important topic that has interest for the larger Earth science community and beyond.

Passchier et al. applies a multi proxy approach to infer sediment provenance and processes from a really nice sediment stratigraphy from the Amundsen Sea continental rise.

Based on the data they present and their discussion, I find their conclusions that grounded ice in the Twaites sector of the West Antarctic Ice Sheet retreated during the early Pliocene both robust and compelling.

Re: We thank Reviewer #1 for this positive evaluation of our work.

While the figures and text are generally clearly presented, I feel that the manuscript would have benefitted from illustrations of the processes behind the different sedimentary facies encountered in the record, including the modern processes that is referenced several times in the manuscript but not thoroughly described. This would especially benefit non-specialists reading this paper.

Re: We have added a diagram (Figure 6) providing a generalized interpretation of the facies in the sequences.

Some comments on figures: The figures is generally clearly presented and readable, however I miss some things regarding captions and labeling etc:

Fig. 2: The labels referred to in the caption are not on the figure. (from PDF).

Re: Labels (a) through (f) were added, and description for (f) was added to the caption.

Fig. 4: Units of the a^* and mSS panels are missing (I suppose the mSS should be μm).

Re: (Now Figure 3). The unit μm was added and a^ is a green-red spectral ratio and does not carry a unit.*

Fig. 5: the a^* records lacks scale (where are the low values..).

Re: included green and red labels and modified the text.

Fig. 8: lowermost panel is not mentioned in the caption, it would also be useful to label the panels a)-C)

Re: (Now Figure 9). The Zr/Hf panel was removed as it was not referred to specifically in the text either. The linear sedimentation rate was added; the caption was modified to reflect these changes.

Fig. 9: labels a)-h) are missing in the figure

Re: labels a) - j) were added to the figure (now Figure 10).

Reviewer #2 (Remarks to the Author):

The manuscript investigates cored materials from the Amundsen Sea in the Pliocene to explore the coeval evolution of the West Antarctic Ice Sheet and oceanographic conditions. The methods presented are robust (though some of the geochemical methods I confess to only having a working understanding of rather than a detailed knowledge) and the inferences from the results are fair and well argued.

In reality, with the caveat of my gap in knowledge of the geochemical methods, this was a very easy paper to review and I had very little to negatively comment upon in the manuscript. It appears to be in a very mature condition, it is well written, the figures help to make the case, and throughout the context is well referenced.

Re: We appreciate these encouraging comments by Reviewer #2.

Given the potential analogy of this warm period the authors investigate, I would state that I think the paper provides an important contribution - though the authors could perhaps consider elaborating on this further. I have some minor suggestions in the pdf.

Re: We now elaborated more on the major findings by framing the results from the Pliocene within a present-day context. We also took note of the minor editorial suggestions in the PDF and modified the text accordingly.

Reviewer #3 (Remarks to the Author):

The authors present an interesting manuscript aimed at understanding the glacial history of the West Antarctic Ice Sheet (WAIS) by examining sediment archives from the deep-sea Antarctic Continental Rise, located downstream from the "doomsday" glaciers of Pine Island and Thwaites. Their goal is to use various proxies (color, grain size, XRF, $^{40}\text{Ar}/^{39}\text{Ar}$) to interpret the extent and dynamics of different WAIS glaciers in relation to Pliocene climate and oceanography.

The study design is solid, the methods align well with the objectives, and I did not find major flaws in the data interpretation.

Re: We are grateful to Reviewer #3 for these positive comments.

However, after multiple readings, I believe several significant issues need to be addressed before the manuscript can be considered for publication.

Below, I provide major comments to help the authors enhance their manuscript, followed by specific feedback in the marked-up version.

Major comment 1:

The manuscript lacks a clearly articulated research question or problem, which is reflected in the overall structure (see also Major Comments 2 and 3). The general aim appears to be an overview of WAIS evolution, primarily focusing on Pine Island and Thwaites glaciers during the Pliocene, to provide analogs of past warm periods and help constrain sea-level rise projections. However, sea-level only appears in the introduction and conclusion, and it's unclear whether the observed retreat

from 4.6 to 4.4 Ma was more or less significant than today's retreat. The manuscript would benefit from a stronger comparison with present-day conditions to help readers grasp the research implications. A clearer focus on the broader problem, followed by the knowledge gap this study aims to address, would enhance the paper's impact. Additionally, the introduction could be better structured: start with the broader problem, state the knowledge gap, and present the methodology, followed by the regional setting and previous work. Much of the material currently in the methods and results sections, such as data from the IODP report and bedrock geology, belongs in this section. Also, geochemical provenance is barely mentioned in the conclusions, despite its focus in the methods

Re: We restructured the manuscript to better reflect how our findings build on existing knowledge and provided the background on the regional setting and geology in the introduction.

Major comment 2:

The manuscript would greatly benefit from a more detailed results section. It is crucial to differentiate between what this study accomplished and what was published during IODP Expedition 379. The authors rely on previously interpreted lithofacies and use them as templates for sedimentation variations but present minimal new results. There are very few numbers or statistics provided, which makes it difficult to understand the dataset or assess the authors' interpretations. I recommend presenting the results from each analysis, such as mean element content by lithofacies. Currently, it feels like the results were skipped, and the manuscript jumps straight into a lengthy discussion.

Re: The notion that "minimal new results" are presented is based on a misunderstanding of shipboard data collection. For example, all particle size data for the facies were collected post-cruise at Montclair State University, as we state in the methods section, and were not generated during the expedition, so they were not previously published in the IODP Expedition 379 Proceedings volume. The same applies to the clay mineral data, which were all analyzed post-cruise at the University of Leipzig (only preliminary shipboard clay-mineral data had been presented in the IODP 379 Proceedings). Additionally, the lower Pliocene lithofacies sequence presented in original Figure 4 (now Figure 3) and Supplemental Figure 2a of our study was also not previously defined and interpreted; instead, in the IODP 379 Proceedings volume only individual lithofacies and two short upper Pleistocene sections with facies associations were described and interpreted without stratigraphic context. Lastly, we present the results for each analysis in various graphic ways, which also display the variations in geochemical ratios by lithofacies. To further clarify the presentation of the data, we added 95% confidence loops to the MDS plot, as suggested by the reviewer in the PDF. Following the referee's recommendation, we have now separated the results section entirely from the discussion section.

Major comment 3:

The discussion would benefit first from an overview of the chronology/the sequence. As of now it is hard to follow track of what happened in chronological order.

You can start by a sub-section of discussion that states what that based on your results, the lithofacies 1 (terrigenous) exhibit characteristics a, b, c. Characteristic a can be associated with ,

characteristic b, can be associated with...etc. The authors kind of do a very brief version of that in the section "Interpretation of sedimentary facies", but I think it is too short– from the text it mostly/solely is: terrigenous facies is interpreted as glacial conditions based on the fact that it is terrigenous; color-banded is associated with the deglacial phase and ice-stream activation based on the high rates of sedimentation; and, the greenish facies are interpreted as interglacial conditions based on low Al/Ti ratios.

After you have interpreted the facies in terms of what they mean in general conditions, present the sequence conditions: based on these facies you have 19 changes (A series) in background conditions that can be tied to LRD04 benthic stack. After correlating tie-points you have fewer A series than glacial-interglacial cycles as suggested by benthic stacks. Discuss why, and if it means something about the WAIS sensitivity to orbital forcing + here you can introduce the variability of in the different A series in chronological order (IRD provenance, sedimentation rates, terrigenous sand content, kaolinite content, etc.) This will set the framework for the subsequent interpretations of changes in the cores.

Re: The detailed facies description and interpretation was moved to the results section. Many details of the facies description and interpretation were previously woven into the discussion. The discussion now includes a chronological order of events, as suggested by the reviewer.

Minor comments:

I have provided a marked-up Word version of the manuscript with detailed comments to make the revision process easier for the authors. This includes direct suggestions and queries.

Comments on figures:

Figure 1:

This figure is difficult to read due to the numerous black lines. Consider dividing it into two panels: one showing topography/bathymetry (with color), glacier extent, and catchments; and another showing geology and Zr/Hf values. Ensure that the colors in the legend match the map for clarity. Something like this with catchments on them and the age of each geological unit:

Re: We changed the inset figure to one with catchments. We provided clarity on the black lines in the legend. We did not expand to a full geological map in addition to the outcrop and rock and sediment sample data our map is based on, because it would include controversial interpretations of geophysical data beyond the scope of this paper. We did add information about the assumed Thwaites Glacier sub-ice geology as inferred from aerogeophysical data (Jordan et al., 2023).

Figure 2:

The figure would be more effective if it showed a transition from glacial to interglacial (dark gray to color-banded to green), rather than starting with color-banded. Also, arrange the cores and history from oldest to youngest, left to right, to match the natural reading flow. Aligning all cores horizontally would make it easier to add curves and improve readability.

Re: Stratigraphically, the color-banded facies is the transition from glacial to interglacial. Showing the continuous XRF data for all three facies in the detailed view of one core section is only possible,

if the thinnest greenish grey facies is the center unit in the view. Therefore, we selected the section shown in Figure 2. We furthermore point out that the natural flow of reading is not universal, and for many paleoclimatologists it is right to left, old to young, which is our consistent choice throughout the manuscript (on an x-axis of a mathematical visualization, zero is the origin on the left and values increase to the right). To accommodate those reading time series from left to right, we have now added “old” and “young” labels in the figures to guide the observations of the core sections and data. Just to give one example: The frequently cited paleoclimate paper about the LR04 benthic stack also displays oldest data on the right and youngest data on the left (Lisiecki and Raymo, 2005; <https://doi.org/10.1029/2004PA001071>).

Figure 3:

Add transparency and outlines to the facies circles to ensure no data points are hidden by others.

Re: Many data points plot on top of each other in this figure (now Figure 4) because they have similar values. However, we carefully chose the symbols and hope that all individual data points are recognizable now. Furthermore, we have added 95% confidence ellipses to the MDS as suggested by Reviewer #3 in their comments on the PDF.

Figure 4:

Include both cores and all curves referenced in the text. Consider removing the core number and recovery information, as these can be found in the IODP report. Align the lithology box with the analyzed interval and ensure the lithology reflects the facies discussed in the text. An idea would be to add the lithology color scheme to the whole figure/under the curves. This would really help with linking proxies and facies.

Re: In the revised paper we present the downcore data plot for Site U1533 in Figure 3 and that for Site U1532 in Supplemental Figure 5. We prefer not to add the Site U1532 down-core data plot to the main manuscript as it would either overload Figure 3 or add an additional figure, which seems unnecessary. We now highlight the greenish grey interglacial beds with green shading under the graphs in the revised Figure 3. The lithological log is based on visual information and smear slides, and provides an overview of the sedimentology, which can be used to characterize the depositional setting and processes of the sediments. The analytical datasets are strongly governed by the provenance of the supplied terrigenous detritus, which is not necessarily the same for identical facies in the stratigraphic column because of changes in maximum and minimum ice-sheet extent through different glacial-interglacial cycles.

Figure 5:

Align the start and end points of the three curves and orient the figure horizontally.

Re: In this paper, drill-site data on a depth scale are consistently displayed vertically, illustrating the orientation of the layers from which the samples were collected (The only exception is revised Figure 2, where just one core section is displayed). Figure 5 displays the stratigraphy on a depth scale, a vertical cross-section of the geological stacking of layers on top of each other. The depth scale is a spatial scale indicating a position beneath the seafloor, which in the reference frame of a person on the surface of the Earth across a drill site is vertically down, so it is natural to read this

data vertically. Core data on an age scale are displayed horizontally as a time series, because the spatial data is converted to a time series using an age model and no longer a true spatial representation.

Figure 6:

Add all argon-argon histograms from Figure 7. Remove the outline from the bins and use color fill instead to reduce visual clutter.

Re: We now present the histograms of the individual stratigraphic intervals in a separate figure, i.e., Figure 8 in the revised manuscript.

Figure 7:

If the curves are added to Figure 4, this figure may no longer be necessary.

Re: The curves were added to Figure 4 (now Figure 3) and the original Figure 7 was removed, as requested.

Figure 8:

Label each chart with a, b, c, and explain each in the main text.

It's unclear where the data points originate. To improve clarity, I recommend differentiating between each core and deriving separate curves for each. This would help determine whether distinct trends or narratives emerge from the individual cores, which isn't evident in the current presentation.

Re: Labels a-c were added to the charts as requested and the interpretation for each panel is given in the figure caption (now Figure 9). Data from the different sites are now marked by different symbols to show that there is overlap between the data from the two sites, and that no trends emerge from either site.

Figure 9:

Excellent figure overall. However, I suggest flipping it to read from left to right, with the oldest data on the left. The gray area of core 32 is a bit lost and could just be in another color. You could present all curves that relates to core 33 in black and to core 32 in blue, in this figure and in all others. I would also add benthic stack at the top or bottom, it is the reference of the study.

Re: We added the benthic stack at the top of the figure (now Figure 10). As mentioned above, the original reference for the benthic stack (Lisiecki and Raymo, 2005) – and also newer versions (see Ahn et al. (2017) – consistently displays the oldest data on the right, as in the submitted version of our manuscript. Therefore, we decided to keep the orientation as is. We did not change the grey areas as we think this visualization of the U1532 data works well.

REVIEWER COMMENTS

Reviewer #1 (Remarks to the Author):

Dear Passchier et al.,

I am satisfied with the reply to the reviewers comments and the changes made to the manuscript.

I recommend publication.

Re: We thank the reviewer for the positive recommendation.

Reviewer #3 (Remarks to the Author):

I appreciate the efforts the authors have made in revising the manuscript. Overall, I find that the adjustments to the structure and the incorporation of suggestions have improved the readability and clarity of the paper. The manuscript is now more cohesive and better organized.

That said, I believe the introduction remains too long (currently spanning five pages) and contains some redundancies, particularly in the statement of the study's objectives, which are repeated multiple times (see marked-up copy).

Re: We agree that after moving the discussion of all previously published data to the introduction, that the introduction is now too long. We shortened the introduction by moving the detailed description of the experimental approach and proxies to the Methods section. In addition, we moved the interpretation of published bedrock trace element data and the review of published Ar/Ar ages of marine sediments back to the discussion section. In the present annotated file the reviewer marked as redundant: "In the light of modern major ice loss in the Amundsen Sea drainage sector of WAIS, we targeted a sedimentary paleoarchive offshore from this sector to reconstruct ice-sheet variability through the warm early Pliocene. Here we use the sedimentological, geochemical and clay mineralogical data collected on the drillcores from the Amundsen Sea continental rise to determine the extent of WAIS retreat during the early Pliocene warm period, and to characterize the mechanisms and impacts of this ice loss." The only sections marked as repetition were: "we aim to reconstruct the connections between source-to-sink supply of glaciogenic detritus and WAIS dynamics" and "the extent in changes in grounding line position within the context of the changing local paleoceanography", which to us do not seem to be repetitions, rather clarifications.

Additionally, I suggest that the last paragraph of the introduction be moved to the discussion, conclusions, or abstract, as it reveals key findings that should not be preemptively disclosed.

Re: The formatting instructions for Nature Communications ask for the inclusion of this section in the Introduction. The formatting instructions state: *The final paragraph must begin with a phrase like “In this work” or “Here, we show”, and contain a brief summary of the major results and conclusions of the current work, written in the present tense.* The formatting instructions also call for the abstract to be no longer than 150 words, and the main text not to include a Conclusion section.

At this stage, my primary concerns are editorial in nature, and I have provided detailed suggestions in the marked-up copy. While I appreciate the inclusion of the new figure illustrating processes, I find that the revisions to the other figures were somewhat minimal. Rather than fully engaging with the constructive feedback, the authors have justified retaining their original figures and made only minor modifications. While these choices do not invalidate the study’s conclusions, I believe they represent missed opportunities to enhance clarity and effectively communicate key results. Per example, I suggested that the core proxies be presented from left to right, progressing from older to younger, as this follows the natural reading order in English and aligns with how information is typically processed in an English-language journal. Since the authors are presenting values in years before present (ie., negative values), the use of a standard x-axis convention—where zero is positioned on the right—would be more intuitive. Adjusting these aspects would improve clarity and better align with standard conventions for chronological data presentation.

Re: In a survey of geology journals with a broad audience, Reshnik et al. (2024) found that nearly 55% of authors used the right-to-left time scale, whereas only 20% used the left-to-right time scale. While this may be counter-intuitive for some, most other authors publishing on Pliocene paleoclimate data work with a right-to-left timescale: Lisiecki and Raymo in *Paleoceanography* (2005), McKay et al. in *PNAS* (2012), Federov et al. in *Nature Geoscience* (2013), Tierney et al. in *GRL* (2019), Dumitru et al. in *Nature* (2019). We argue that the convention in displaying paleoclimate data time series for the Pliocene is from right to left, and that users of Pliocene paleoclimate data would prefer to read all data series in the same direction.

Figures should prioritize the most essential information to support the paper’s arguments. Currently, many figures feel overcrowded or could be optimized for readability. Below, I highlight some specific concerns:

Figure 1: The inclusion of hillshade beneath the ice introduces visual noise, making it difficult to discern key features. Consider whether this layer is necessary. Additionally, the figure contains many overlapping black lines, reducing readability. A possible solution would be to split the figure into two panels: one focusing on topography/bathymetry (with glacier extent and catchments) and another dedicated to geology and Zr/Hf values. Ensuring that legend colors match the map would also improve clarity.

Re: We have now outlined the catchments in white, which eliminates the crossing of black lines. Our study discusses the evacuation of rock materials from the different catchments, so the catchment lines are necessary. Both the basement topography or “hillshade” and the Zr/Hf values are data on the bedrock geology, and they should occur together in the same figure panel for clarity and not in separate panels. Our approach is similar to Marschalek et al. 2021, published in Nature, who provide a “hillshade” type of base map for a Miocene provenance study. We improved the consistency of the legend colors.

Figure 3: The resolution of the curves appears low or blurry. Additionally, the alignment of lithology, magnetic susceptibility, and core recovery should match the start of the curves for consistency.

Re: The lithology and data curves are now aligned. In response to the comment about figures being overcrowded, we have now removed the Sm/Zr panel, because it was only mentioned once in the results and was not specifically mentioned in the interpretation. The blurriness resulted from the conversion to PDF during production of the manuscript file.

Figure 5: The start and end points of the three curves should be aligned. It is unclear why the benthic stack appears to “float” on the left.

Re: We have moved the benthic stack to the right of the curves and lined out the top of the benthic stack with the downhole curve for Site U1533. The benthic stack is on an age scale, whereas the data is on a depth scale and because of the variable sedimentation rates, there is no one-to-one correlation possible.

Figures 7 and 8: The current bin outlines add visual clutter. I recommend either removing the outlines and using color fills instead or add color to the curves to enhance contrast and improve readability.

Re: We removed the outlines and added a grey color to the bins.

While these are not fundamental issues, I believe addressing them would substantially improve the manuscript’s presentation and accessibility to readers. I encourage the authors to consider these refinements to enhance the overall clarity of their figures.

Review of "West Antarctic ice retreat and deepwater formation in the Amundsen Sea in the warm early Pliocene" by Passchier et al.

Comments to editor

5 The authors present an interesting manuscript aimed at understanding the glacial history of the West Antarctic Ice Sheet (WAIS) by examining sediment archives from the deep-sea Antarctic Continental Rise, located downstream from the "doomsday" glaciers of Pine Island and Thwaites. Their goal is
10 to use various proxies (color, grain size, XRF, $40\text{Ar}/39\text{Ar}$) to interpret the extent and dynamics of different WAIS glaciers in relation to Pliocene climate and oceanography.

The study design is solid, the methods align well with the objectives, and I did not find major flaws in the data
15 interpretation. However, after multiple readings, I believe several significant issues need to be addressed before the manuscript can be considered for publication. Below, I provide major comments to help the authors enhance their manuscript, followed by specific feedback in the marked-up
20 version. Comments to the authors:

Major comment 1:

The manuscript lacks a clearly articulated research question or problem, which is reflected in the overall structure (see also Major Comments 2 and 3). The general aim appears to be an
25 overview of WAIS evolution, primarily focusing on Pine Island and Thwaites glaciers during the Pliocene, to provide analogs of past warm periods and help constrain sea-level rise projections. However, sea-level only appears in the introduction and conclusion, and it's unclear whether the
30 observed retreat from 4.6 to 4.4 Ma was more or less significant than today's retreat. The manuscript would benefit from a stronger comparison with present-day conditions to help readers grasp the research implications. A clearer focus on the broader problem, followed by the knowledge gap this
35 study aims to address, would enhance the paper's impact. Additionally, the introduction could be better structured: start with the broader problem, state the knowledge gap, and present the methodology, followed by the regional setting and previous work. Much of the material currently in the methods
40 and results sections, such as data from the IODP report and

bedrock geology, belongs in this section. Also, geochemical provenance is barely mentioned in the conclusions, despite its focus in the methods

Major comment 2:

The manuscript would greatly benefit from a more detailed results section. It is crucial to differentiate between what this study accomplished and what was published during IODP Expedition 379. The authors rely on previously interpreted lithofacies and use them as templates for sedimentation variations but present minimal new results. There are very few numbers or statistics provided, which makes it difficult to understand the dataset or assess the authors' interpretations. I recommend presenting the results from each analysis, such as mean element content by lithofacies. Currently, it feels like the results were skipped, and the manuscript jumps straight into a lengthy discussion.

Major comment 3:

The discussion would benefit first from an overview of the chronology/the sequence. As of now it is hard to follow track of what happened in chronological order.

You can start by a sub-section of discussion that states what that based on your results, the lithofacies 1 (terrigenous) exhibit characteristics a, b, c. Characteristic a can be associated with , characteristic b, can be associated with...etc. The authors kind of do a very brief version of that in the section "Interpretation of sedimentary facies", but I think it is too short – from the text it mostly/solely is: terrigenous facies is interpreted as glacial conditions based on the fact that it is terrigenous; color-banded is associated with the deglacial phase and ice-stream activation based on the high rates of sedimentation; and, the greenish facies are interpreted as interglacial conditions based on low Al/Ti ratios.

After you have interpreted the facies in term of what they mean in general conditions, present the sequence

conditions : based on these facies you have 19 changes (A series) in background conditions that can be tied to LRD04 benthic stack. After correlating tie-points you have fewer series than glacial-interglacial cycles as suggested by benthic stacks. Discuss why, and if it means something about the WAIS sensibility to orbital forcing + here you can introduce the variability of in the different A series in chronological order (IRD provenance, sedimentation rates, terrigenous sand content, kaolinite content, etc.) This will set the framework for the subsequent interpretations of changes in the cores.

Minor comments:

I have provided a marked-up Word version of the manuscript with detailed comments to make the revision process easier for the authors. This includes direct suggestions and queries.

Comments on figures:

Figure 1:

This figure is difficult to read due to the numerous black lines. Consider dividing it into two panels: one showing topography/bathymetry (with color), glacier extent, and catchments; and another showing geology and Zr/Hf values. Ensure that the colors in the legend match the map for clarity. Something like this with catchments on them and the age of each geological unit:

REDACTED

Figure 2:

The figure would be more effective if it showed a transition from glacial to interglacial (dark gray to color-banded to green), rather than starting with color-banded. Also, arrange the cores and history from oldest to youngest, left to right, to match the natural reading flow. Aligning all cores horizontally would make it easier to add curves and improve readability.

Figure 3:

Add transparency and outlines to the facies circles to ensure no data points are hidden by others.

Figure 4:

Include both cores and all curves referenced in the text. Consider removing the core number and recovery information, as these can be found in the IODP report. Align the lithology box with the analyzed interval and ensure the lithology reflects the facies discussed in the text. An idea would be to add the lithology color scheme to the whole figure/under the curves. This would really help with linking proxies and facies.

Figure 5:

Align the start and end points of the three curves and orient the figure horizontally.

Figure 6:

Add all argon-argon histograms from Figure 7. Remove the outline from the bins and use color fill instead to reduce visual clutter.

Figure 7:

If the curves are added to Figure 4, this figure may no longer be necessary.

Figure 8:

Label each chart with a, b, c, and explain each in the main text. It's unclear where the data points originate. To improve clarity, I recommend differentiating between each core and deriving separate curves for each. This would help determine whether distinct trends or narratives emerge from the individual cores, which isn't evident in the current presentation.

Figure 9:

Excellent figure overall. However, I suggest flipping it to read from left to right, with the oldest data on the left. The gray area of core 32 is a bit lost and could just be in another color. You could present all curves that relates to core 33 in black and to core 32 in blue, in this figure and in all others. I would also add benthic stack at the top or bottom, it is the reference of the study.

Review of "West Antarctic ice retreat and deepwater formation in the Amundsen Sea in the warm early Pliocene" by Passchier et al.

Comments to editor

5 I appreciate the efforts the authors have made in revising the manuscript. Overall, I find that the adjustments to the structure and the incorporation of suggestions have improved the readability and clarity of the paper. The manuscript is now more cohesive and better organized.

10 That said, I believe the introduction remains too long (currently spanning five pages) and contains some redundancies, particularly in the statement of the study's objectives, which are repeated multiple times (see marked-up copy). Additionally, I suggest that the last paragraph of the introduction be moved to the discussion, conclusions, or abstract, as it reveals key findings that should not be preemptively disclosed.

At this stage, my primary concerns are editorial in nature, and I have provided detailed suggestions in the marked-up copy.

20 While I appreciate the inclusion of the new figure illustrating processes, I find that the revisions to the other figures were somewhat minimal. Rather than fully engaging with the constructive feedback, the authors have justified retaining their original figures and made only minor modifications.

25 While these choices do not invalidate the study's conclusions, I believe they represent missed opportunities to enhance clarity and effectively communicate key results. Per example, I suggested that the core proxies be presented from left to right, progressing from older to younger, as this follows the natural reading order in English and aligns with how information is typically processed in an English-language journal. Since the authors are presenting values in years before present (ie., negative values), the use of a standard x-axis convention—where zero is positioned on the right—would be more intuitive. Adjusting these aspects would improve clarity and better align with standard conventions for chronological data presentation.

Figures should prioritize the most essential information to support the paper's arguments. Currently, many figures feel

overcrowded or could be optimized for readability. Below, I highlight some specific concerns:

Figure 1: The inclusion of hillshade beneath the ice introduces visual noise, making it difficult to discern key features. Consider whether this layer is necessary. Additionally, the figure contains many overlapping black lines, reducing readability. A possible solution would be to split the figure into two panels: one focusing on topography/bathymetry (with glacier extent and catchments) and another dedicated to geology and Zr/Hf values. Ensuring that legend colors match the map would also improve clarity.

REDACTED

Figure 3: The resolution of the curves appears low or blurry. Additionally, the alignment of lithology, magnetic susceptibility, and core recovery should match the start of the curves for consistency.

Figure 5: The start and end points of the three curves should be aligned. It is unclear why the benthic stack appears to "float" on the left.

Figures 7 and 8: The current bin outlines add visual clutter. I recommend either removing the outlines and using color fills instead or add color to the curves to enhance contrast and improve readability.

While these are not fundamental issues, I believe addressing them would substantially improve the manuscript's presentation and accessibility to readers. I encourage the authors to consider these refinements to enhance the overall clarity of their figures.